# Ribosome impairment regulates intestinal stem cell identity via ZAKα activation

Joana Silva ®[1], Ferhat Alkan ®[1], Sofia Ramalho[1], Goda Snieckute[2], Stefan Prekovic ®[1], Ana Krotenberg Garcia[3], Santiago Hernández-Pérez[1], Rob van der Kammen[1], Danielle Barnum[1], Liesbeth Hoekman[4], Maarten Altelaar[4,5], Wilbert Zwart ®[1], Saskia Jacoba Elisabeth Suijkerbuijk ®[3], Simon Bekker-Jensen ®[2] & William James Faller ®[1] ✉

The small intestine is a rapidly proliferating organ that is maintained by a small population of *Lgr5*-expressing intestinal stem cells (ISCs). However, several *Lgr5*-negative ISC populations have been identified, and this remarkable plasticity allows the intestine to rapidly respond to both the local environment and to damage. However, the mediators of such plasticity are still largely unknown. Using intestinal organoids and mouse models, we show that upon ribosome impairment (driven by *Rptor* deletion, amino acid starvation, or low dose cyclohexamide treatment) ISCs gain an *Lgr5*-negative, fetal-like identity. This is accompanied by a rewiring of metabolism. Our findings suggest that the ribosome can act as a sensor of nutrient availability, allowing ISCs to respond to the local nutrient environment. Mechanistically, we show that this phenotype requires the activation of ZAKα, which in turn activates YAP, via SRC. Together, our data reveals a central role for ribosome dynamics in intestinal stem cells, and identify the activation of ZAKα as a critical mediator of stem cell identity.

The mammalian intestine is an extremely proliferative organ. The entire epithelium turns over in around 3–5 days, and this proliferation is maintained by cells within the intestinal crypt. The crypt is home to a well-defined stem cell niche that supports a population of rapidly cycling ISCs, found at its base[1]. These stem cells express *Lgr5*[2], and are reliant on WNT ligands that are produced by various cells in the intestine, primarily the Paneth cell and various stromal cells[3,4]. It is known however, that the *Lgr5*-positive population is dispensable for normal intestinal growth[5], and that multiple cell types with stem cell capacity exist in the intestine, including those marked by MEX3A, mTERT, LRIG1, HOPX, DCLK1, and DLL1[6]. For example, ALPI, a marker of the enterocyte lineage, labels cells that normally migrate up the intestinal villus and are shed into the lumen within days[7]. Upon the loss

of *Lgr5*-positive cells however, these cells can undergo dedifferentiation into stem cells, demonstrating the remarkable plasticity of the organ[8]. More recently, a population of cells that appear to regain markers of the fetal intestine has been identified[9], and has been shown to be crucial in a number of contexts, including after helminth infection[10], and extracellular matrix remodeling[11,12]. They have also been shown to be a colitis-associated ISC that contributes to mucosal repair in this condition[13,14].

While these various populations have been characterized, there is still a lack of understanding of the specific signaling pathways that determine alternative ISC identity. The canonical *Lgr5*-positive ISC is well studied, and the role of WNT, EGF, Notch, Bmp, and Hedgehog have been well defined (reviewed in ref. 15). This is not the case for

[1]Division of Oncogenomics, The Netherlands Cancer Institute, Amsterdam, The Netherlands. [2]Center for Healthy Aging, Department of Cellular and Molecular Medicine, University of Copenhagen, Copenhagen, Denmark. [3]Institute of Byodynamics and Biocomplexity, Department of Biology, Faculty of Science, Utrecht University, Utrecht, The Netherlands. [4]Proteomics Facility, The Netherlands Cancer Institute, Amsterdam, The Netherlands. [5]Biomolecular Mass Spectrometry and Proteomics, Bijvoet Center for Biomolecular Research, Utrecht Institute for Pharmaceutical Sciences, Utrecht University and Netherlands Proteomic Centre, Utrecht, The Netherlands. ✉e-mail: w.faller@nki.nl

other ISC types however, and although some physical determinants of ISC identity have been identified[12,16], the signaling cascades that define various phenotypes are largely unknown.

In several organs, mRNA translation has been shown to play a key role in stem cell fate[17,18]. In hematopoiesis for example, simply increasing or decreasing protein synthesis in stem cells by 30% is sufficient to completely alter cell fate decisions[19]. In the intestine, altered translation is known to be crucial for intestinal tumorigenesis[20-22], and while several studies have demonstrated the importance of mTOR signaling in ISCs (particularly under stress conditions)[23-25], there are relatively few studies that directly analyze the role of translation in these cells[26].

In this study, we observe that the ISC is very responsive to the inhibition of RNA translation. Following a decrease in protein synthesis we see a loss of *Lgr5*-positivity and the induction of a fetal-like phenotype. This is accompanied by a translationally regulated switch in the metabolic identity of the cells, from oxidative phosphorylation to glycolysis. We show that this phenotype switch is driven by ribosome impairment. Although such translational issues have been shown to be a common occurrence[27-29], their biological relevance is currently unknown. Here, we provide insights into a biologically relevant change in cell fate caused by such impairment. Furthermore, we show that this phenotypic switch occurs in a ZAKα-dependent manner, via activation of SRC and YAP, and that amino acid restriction results in the same ISC-identity change both in vitro and in vivo. We thus show that ribosome dynamics regulate intestinal cell fate, and may act as a sensor of nutrient availability, allowing the intestine to adapt to low amino acid levels by altering ISC identity.

## Results

### Translation inhibition causes a cell and metabolic identity switch in the ISC

In order to examine the role of RNA translation in the ISC, we generated organoids from the intestines of *VilCre*[ERT2]*;Rptor*[fl/fl] mice[30-32] (Supp. Fig. 1a). Raptor is an essential component of mTORC1, and we have previously shown that this complex regulates RNA translation in WNT-high intestinal cells[20]. As expected, upon tamoxifen induction there was a loss of Raptor and mTORC1 signaling (as measured by S6 phosphorylation), and [35]S-methionine incorporation showed that these organoids had decreased levels of protein synthesis (Fig. 1a, b). Surprisingly, the organoids took on a strikingly different morphology, losing the expected budding phenotype, and becoming cystic (Fig. 1c), which was reminiscent of organoids that were enriched for stem cells[33]. However, qPCR analysis revealed that the organoids did not express ISC markers (*Lgr5*, *Mex3a* and *Axin2*), but instead were enriched for markers commonly associated with the fetal intestine (*Tacstd2*, *Sca1*, *Spp1* and *Cnx43*)[9] (Fig. 1d). These results were further validated by immuno-fluorescence staining, revealing a complete loss of the ISC marker OLFM4 and gain of fetal markers TACSTD2 and SCA1 upon *Rptor* deletion (Fig. 1e). Additionally, we also observed a significant loss of the differentiation markers LYZ1 and ALDOB (Supp. Fig. 1b) in *Rptor*-deficient organoids, supporting the idea that inhibition of translation, through mTORC1 inactivation, affects intestinal cell identity.

This fetal-like ISC population has previously been linked to intestinal damage response[10,11], and has been shown to play a role in oncogenic growth[34]. It has previously been shown that the metabolic identity of ISCs is key in maintaining their *Lgr5*-positive identity, with these cells requiring a high level of oxidative phosphorylation[35]. We therefore carried out Seahorse analysis on these organoids. This analysis showed that *Rptor*-deleted organoids had a reduced level of oxygen consumption, and a higher level of lactate production, suggesting a switch from oxidative phosphorylation to glycolysis (Fig. 1f; Supp. Fig. 1c). It is known that Paneth cells also have a high level of glycolysis, but there is a loss of Paneth cells following *Rptor* deletion, which discounts this as the cause of

the increased glycolysis (Fig. 1d, Supp. Fig. 1b). Surprisingly, while RNA Seq showed the expected activation of a fetal signature[9] and an inhibition of both the *Lgr5*[7] and mTORC1 signature, it did not show an up-regulation of mRNAs associated with glycolysis, and in fact showed a down-regulation of this pathway by GSEA analysis, suggesting that this metabolic change is not transcriptionally regulated (Fig. 1g & Supp. Fig. 2a–c).

As mTOR is a well-known regulator of mRNA translation[36], this is an obvious alternative mechanism by which glycolysis could be regulated. We therefore carried out Ribosome Profiling from *Rptor*[fl/fl] organoids and wild type organoids that had been enriched for adult ISCs, using established culture conditions[33] (Supp. Fig. 3a), allowing us to compare adult ISCs with fetal-like ISCs. The organoids also expressed RPL22 tagged with HA[37] to increase the efficiency of ribosome isolation, and we calculated the translation efficiency (TE) of each mRNA, which is ratio of ribosome binding and RNA abundance as determined by ribosome profiling and RNA-seq[38] (Fig. 2a, b & Supp. Fig. 4a–d). This experiment showed that upon Raptor loss and the induction of a fetal-like stem cell identity, there was an increase in the TE of glycolysis associated mRNAs, and a decrease in the TE of oxidative phosphorylation-related mRNAs (Fig. 2c & Supp. Fig. 3b), confirming the translational regulation of this process.

Alongside this, we generated *Lgr5Cre*[ERT2]*;Rptor*[fl/fl]*;RiboTag*[HA/HA] animals[2]. This allowed the in vivo deletion of *Rptor* and the inducible expression of the RiboTag allele specifically within the *Lgr5*-positive ISC population (Fig. 2d). Twenty four hours after tamoxifen induction, the RiboTag[HA/HA] allele was expressed exclusively in the crypt base, and could be used to isolate ribosomes specifically from these cells (Fig. 2e). We could then use this material to carry out an in vivo ribosome profiling experiment, comparing *Lgr5Cre*[ERT2]*;RiboTag*[HA/HA] with *Lgr5Cre*[ERT2]*;Rptor*[fl/fl]*;RiboTag*[HA/HA] animals, thus measuring the impact of *Rptor* deletion specifically within the ISC in vivo (Supp. Fig. 5a–e). As we observed in organoids, deletion of *Rptor* in vivo results in the upregulation of the fetal signature, and an increase in the ribosome association of mRNAs associated with glycolysis (Fig. 2f), showing that the same process occurs in vivo.

### Ribosome impairment causes the adult to fetal-like ISC identity switch

Upon further analysis of this data, it was observed that there was a striking increase in ribosome occupancy specifically on stop codons following *Rptor* deletion, suggestive of stalling ribosomes on these codons (Fig. 2g & Supp. Fig. 6). Such stalling (and subsequent ribosome collisions) around stop codons has been observed previously[29], however the biological relevance of this is unknown. Stalled ribosomes are known to have wide-ranging effects on translation dynamics, via recruitment of the ribosome quality control machinery[39], activation of GCN2[40], and an increase in ribosome collisions[41,42].

As both stalled and collided ribosomes are known to activate cellular signaling, we assessed the ability of cycloheximide to drive the same adult to fetal-like stem cell switch that we observed following *Rptor* deletion. We used both low (0.015 μg/ml) and high (100 μg/ml) doses of cycloheximide, as low doses inhibit the elongation of some ribosomes, causing ribosome collisions, whereas high doses stall all ribosomes, allowing us to differentiate between the two situations[41,43]. Importantly, both doses result in similar inhibition of protein synthesis (Supp. Fig. 7a). However, exclusively at low doses, cycloheximide treatment phenocopied Raptor loss, resulting in a loss of *Lgr5*-positivity and an increase in markers of fetal-like stem cells (Fig. 3a). This suggests that the ribosome impairment caused by *Rptor* deletion includes an increase in ribosome collisions (particularly around the stop codons), and that this may be causing the observed phenotype.

Ribosome collisions and other impairments of translational elongation are thought to be common events, and are known to have

different outcomes depending on the driver and circumstances of the problem. For example, translation continues after some collisions, while mRNA and nascent peptide degradation occurs after others[44,45]. They are also a natural response to many stimuli, including damaged or misfolded mRNA[46,47], and amino acid depletion[48]. As the intestine is a major nutrient sensing organ, we asked whether translational impairment driven by amino acid depletion could also drive the same alteration in ISC populations. We therefore grew the organoids in media depleted of glutamine or leucine. This resulted in a switch of ISC identity (Fig. 3b), suggesting that changes in ribosome dynamics may act as a nutrient sensor in order to alter stem cell dynamics in response to local nutrient availability.

The most well characterized detector of amino acid availability is GCN2, which is known to recognize uncharged tRNAs, and drive the cellular response to amino acid limitation[49]. GCN2 has also previously been linked to the detection of ribosome stalling, which caused an activation of the integrated stress response and eIF2α phosphorylation[41]. However, *Rptor* deletion, cycloheximide treatment and, most surprisingly, amino acid restriction did not result in

an increase in eIF2α phosphorylation (Supp. Fig. 7b). Furthermore, inhibition of GCN2 was not sufficient to block the stem cell identity switch (Supp. Fig. 7c), suggesting that the canonical nutrient sensing pathways are not used to respond to low levels of amino acids in the ISC, nor is this pathway used to detect ribosome impairment.

**Phenotype switch is regulated via a ZAKα-mediated signaling cascade.** Recent studies have pinpointed the long isoform of the MAPKKK family member ZAK (ZAKα) as a key mediator of cellular responses to ribosome stalling and collisions[41,43,50]. We therefore assessed ZAKα phosphorylation, following *Rptor* deletion, cycloheximide treatment, or glutamine/leucine deprivation. This analysis showed that there was an increase in the phosphorylation of ZAKα in all of these conditions (Fig. 3c). To understand the importance of ZAKα in ISC identity, we depleted it in organoids using shRNA. It is important to note that the shRNA used is specific to the long isoform of ZAK, and the levels of ZAKβ remain unchanged (Supp Fig. 7e). Strikingly, ZAKα inhibition completely blocked the ability of cycloheximide to activate the fetal signature, and the metabolic switch that accompanies it (Fig. 3d, e).

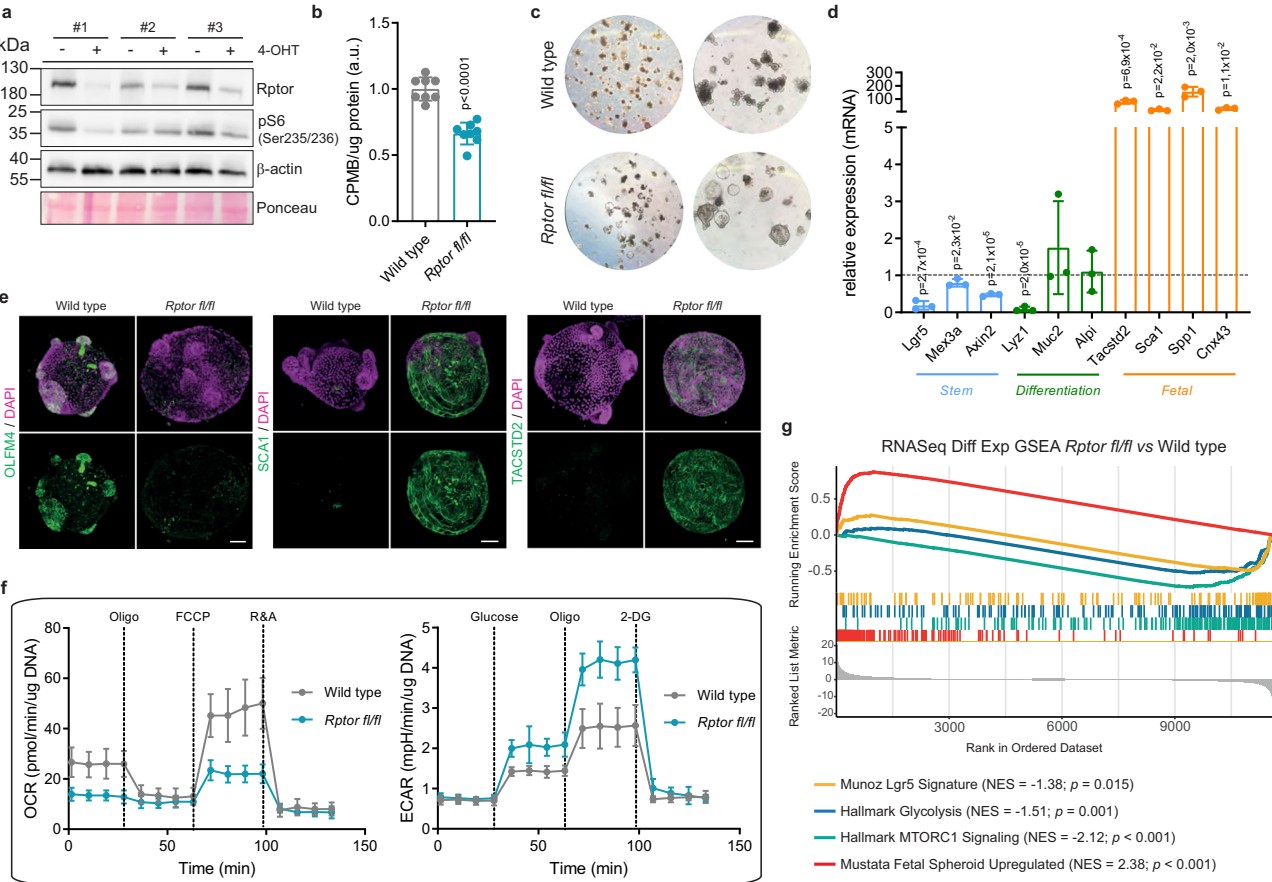

**Fig. 1 | Inhibition of translation drives stem cell identity changes. a** Western blot analysis confirming reduced Raptor and pS6 (Ser235/236) protein levels in cells from 3 different animals treated with 4-Hydroxytamoxifen. β-actin and Ponceau serve as loading control. **b** Incorporation of ³⁵S-methionine shows decreased protein synthesis in *Rptor^fl/fl* organoids compared to WT. Mean and standard deviation are shown (*n* = 4 biological replicates each accessed in technical duplicates). *p* values were determined using a two-tailed *t*-test. **c** Representative images showing morphological differences in organoids upon *Rptor* loss, compared to WT. Pictures were acquired using 5x (left panel) and 20x (right panel) magnifications. **d** RT-qPCR analysis of individual genes related to stem (*Lgr5*, *Mex3a* and *Axin2*), differentiation (*Lyz1*, *Muc2* and *Alpi*), and fetal-like state (*Tacstd2*, *Sca1*, *Spp1* and *Cnx43*) of *Rptor^fl/fl* organoids compared to WT, using *Hprt* as a reference. Mean and standard error of the mean are shown (*n* = 3 biological replicates each accessed in technical

triplicates). *p* values were determined using a two-tailed *t*-test. **e** Representative 3D-reconstructed confocal images of wild type and *Rptor^fl/fl* organoids show significant reduction of adult stem cell marker OLFM4 (left panel, green) and activation of fetal markers SCA1 (middle panel, green) and TACSTD2 (right panel, green). Dapi is used to visualize the nuclei (magenta). Scale bar is 50 μm. **f** OCR and ECAR analyses reveal decreased respiration and increased glycolysis in *Rptor^fl/fl* organoids compared to WT. Mean and standard deviation are shown (*n* = 2 biological replicates each accessed in technical quadriplicates). Refer to Supp. Fig. 1 for quantifications. **g** Gene Set Enrichment Analysis based on RNASeq differential expression data comparing *Rptor^fl/fl* organoids to WT (*n* = 4 from 2 biological replicates for each). Enrichment is shown for transcriptional signatures related to stemness, glycolysis, mTORC1 signaling and fetal-like state. *p* values were determined using the *clusterProfiler* package.

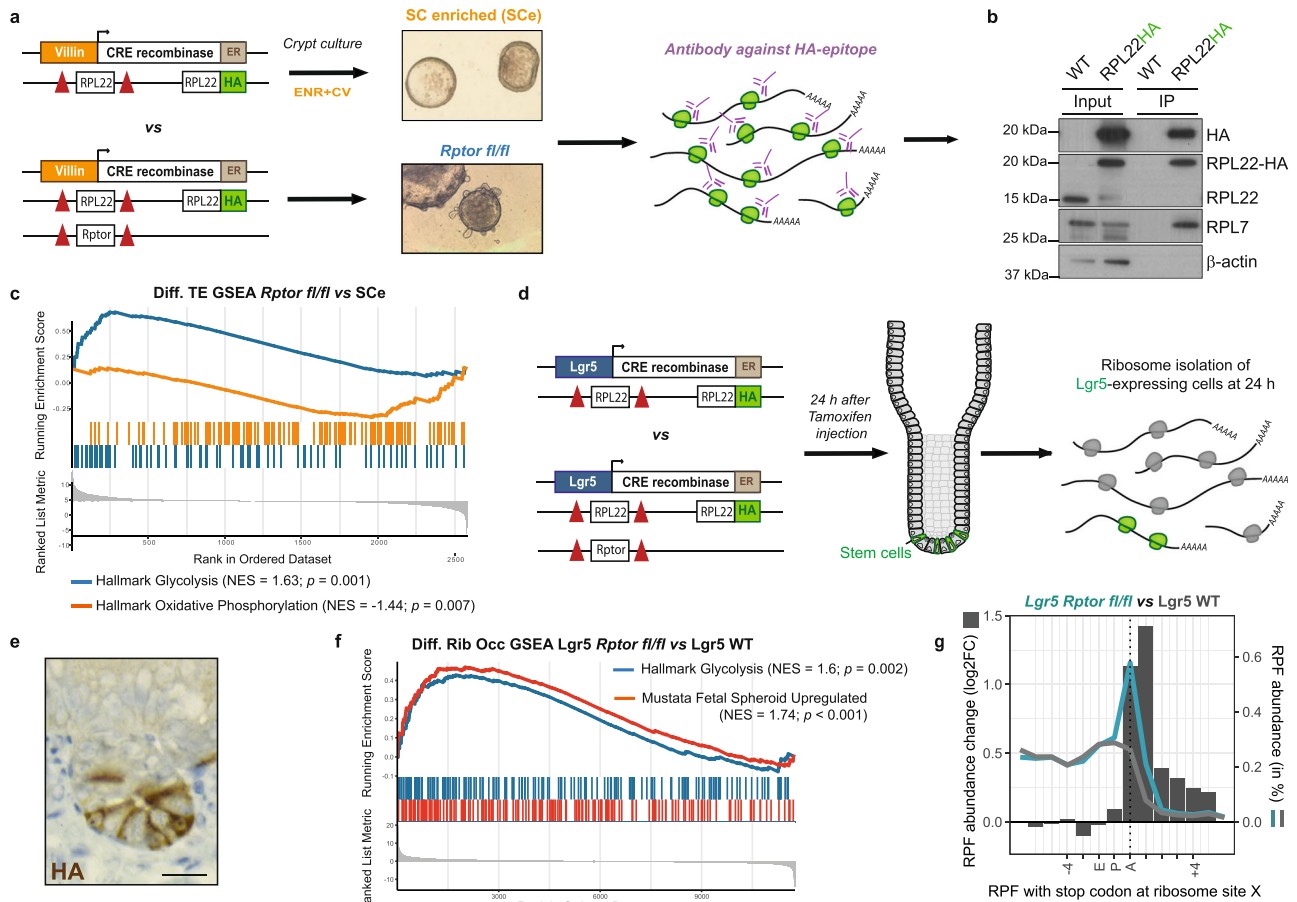

**Fig. 2 | Metabolism is translationally regulated in fetal-like ISCs. a** Experimental workflow of the RiboSeq experiment comparing the translatome of *Rptor*[fl/fl] organoids to WT. Crypt cultures were generated from *VillinCre*[ERT2]RPL22.HA mice both in the presence and absence of Raptor. Tagged ribosomes were pulldown using an HA antibody and ribosome protected fragments were sequenced and mapped to the genome. **b** Western blot analysis confirming efficient pulldown of HA-tagged RPL22 ribosomes. RPL7 co-immunoprecipitation is used as an indicator of intact ribosomes. β-actin serves as a loading control. Experiments were done in one biological replicate. **c** Gene Set Enrichment Analysis based on differential translation efficiency data comparing *Rptor*[fl/fl] organoids to stem cell enriched cultures (*n* = 3). Enrichment is shown for transcriptional signatures related to oxidative phosphorylation and glycolysis. p values were determined using the *clusterProfiler* package. **d** Experimental workflow of the RiboSeq experiment comparing the translatome of intestinal stem cells from Rptor[fl/fl] to WT mice. Ribosomes were tagged specifically in the instestinal stem cells of mice using an *Lgr5Cre*[ERT2]

promoter, both in the presence and absence of Raptor. 24 h after tamoxifen induction, the ribosomes are recombined with the HA tag exclusively in the intestinal stem cells, allowing the capture and further study of their translatome by RiboSeq. **e** Imunohistochemistry staining of Lgr5Cre[ERT2]RPL22.HA intestines show that HA staining is restricted to ISCs. Stainings were done in 3 biological replicates. Scale bar: 20 μm. **f** Gene Set Enrichment Analysis based on differential ribosomal occupancy data comparing the intestinal stem cells of *Rptor*[fl/fl] and WT mice (*n* = 3 biological replicates). Enrichment is shown for transcriptional signatures related to glycolysis and fetal-like state. *p* values were determined using the *clusterProfiler* package. **g** Distribution of RPFs along transcripts show an accumulation of reads in the stop codon in the *Lgr5Cre*[ERT2]*Rptor*[fl/fl]RPL22.HA mice (blue) compared with the Lgr5Cre[ERT2]RPL22.HA (gray). Barplot depicts RPF abundance change (log2FC) between the two conditions and lines show total RPF abundance (%), for which RPFs are grouped based on their A-site position with stop-codon as reference.

This was also the case for *Rptor* deletion, and amino acid restriction (Fig. 3f–h). Additionally, Vemurafenib has been shown to be a potent inhibitor of ZAK[51,52]. As with shRNA mediated silencing of ZAKα, treatment with Vemurafenib also blocked ISC identity switch, as well as the accompanying metabolic switch (Supp. Fig. 8a–d).

As we had already shown that this effect is GCN2-eIF2α independent, we measured the activation of the JNK and p38 signaling pathways, as these are known to mediate the downstream consequences of ZAKα activation[50]. There was no activation of JNK signaling following *Rptor* deletion (Supp. Fig. 7f), and inhibition of this pathway with JNK-IN-8 was unable to block the adult to fetal-like stem cell transition (Supp. Fig. 7g). While we did observe an increase in p38 phosphorylation, supporting the observation of ZAKα activation (Supp. Fig. 7f), inhibition using SB203580 was also unable to block the transition (Supp. Fig. 7g), suggesting that the signaling pathways previously described to be downstream of ZAKα are not responsible for the stem cell identity switch.

We therefore set out to identify other potential substrates of ZAKα, by defining its specific interactome in our fetal-like system. To do this, we carried out immunoprecipitations of FLAG tagged ZAKα in both wild-type and *Rptor*-deficient organoids, followed by mass spectometry (IP-MS) (Supp. Fig. 9a). While we could only identify a small number of proteins that were pulled down with ZAKα in our control cells (Supp. Fig. 9b), Raptor loss caused a dramatic increase in the number of proteins that co-precipitated with ZAKα, highlighting the importance of this kinase in mediating the cellular response to impaired translation (Fig. 4A). Among these interactors we identified SRC, a well known YAP activator which has been recently implicated in injury-induced regeneration in the intestine[11]. In that study, the authors showed that upon tissue damage the intestinal epithelium is converted into a fetal-like state, with decreased levels of differentiation markers, which seems to be mediated by increased FAK/SRC signaling and a resulting YAP/TAZ activation. To validate the interaction of ZAKα with SRC in our other fetal-like models, we performed a Co-IP of ZAKα-FLAG

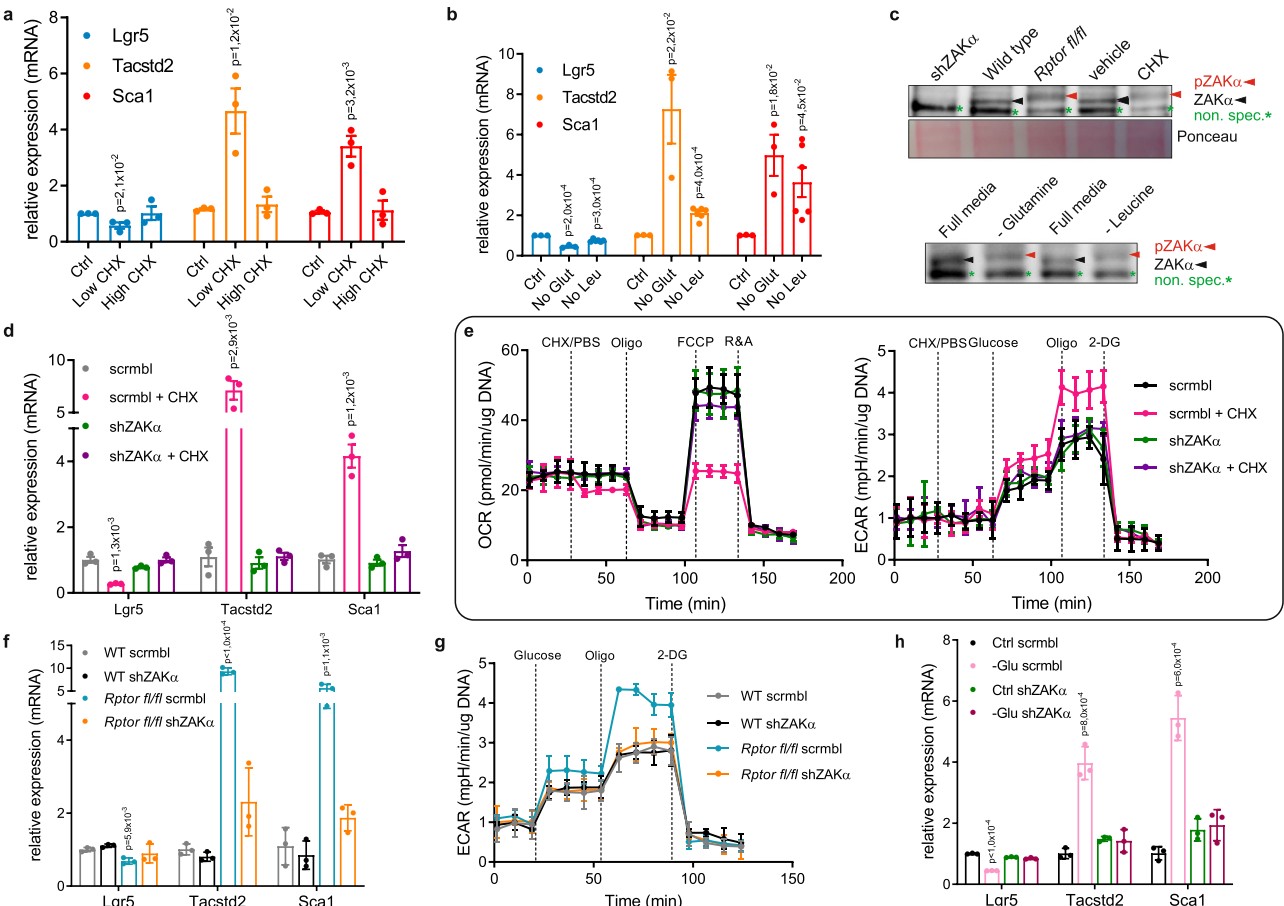

**Fig. 3 | Ribosome impairment drives metabolic changes and the fetal signature via ZAKα activation. a** RT-qPCR analysis of genes related to stem (*Lgr5*) and fetal-like state (*Tacstd2* and *Sca1*) of WT organoids treated with cycloheximide at either a low (0.015 ug/ml) or high dose (100 ug/ml) for 30 min. *Hprt* is used as a reference. Mean and SEM are shown (*n* = 3 biological replicates each accessed in technical triplicates). *p*-values were determined using a two-tailed *t*-test. **b** RT-qPCR analysis of genes related to stem (*Lgr5*) and fetal-like state (*Tacstd2* and *Sca1*) of WT organoids cultured in media without glutamine or leucine for 1 h. *Hprt* is used as a reference. Mean and SEM are shown (*n* = 3 biological replicates each accessed in technical triplicates). *p*-values were determined using a two-tailed *t*-test. **c** Western blot confirming activation of ZAKα in *Rptor*<sup>fl/fl</sup> and low-dose cycloheximide treated organoids (upper panel) and organoids deprived from glutamine and leucine (bottom panel) compared to WT. Red arrows indicate phosphorylated ZAKα, black arrows indicate the ZAKα levels, green stars indicate non-specific bands. Ponceau serves as a loading control. Blots were done on two animals. **d** RT-qPCR analysis of genes related to stem (*Lgr5*) and fetal-like state (*Tacstd2* and *Sca1*) of WT organoids treated with low-dose cycloheximide (0.015 ug/ml) for 30 min in the presence or absence of ZAKα. *Hprt* is used as a reference. Mean and SEM are shown (*n* = 3 biological replicates each accessed in technical triplicates). *p*-values were determined using a two-tailed *t*-test. **e** OCR and ECAR analyses reveal that deleting ZAKα rescues the metabolic changes caused by low-dose cycloheximide (0.015 ug/ml). Mean and SD are shown (*n* = 1 biological replicate accessed in technical quadriplicates). **f** RT-qPCR analysis of genes related to stem (*Lgr5*) and fetal-like state (*Tacstd2* and *Sca1*) of WT and *Rptor*<sup>fl/fl</sup> organoids in the presence or absence of ZAKα. *Hprt* is used as a reference. Mean and SEM are shown (*n* = 1 biological replicate accessed in technical triplicates). *p*-values were determined using a two-tailed *t*-test. **g** ECAR analysis show that ZAKα deletion rescues the metabolic changes caused by *Rptor* deletion. Mean and SD are shown (*n* = 1 biological replicate accessed in technical quadriplicates). **h** RT-qPCR analysis of genes related to stem (*Lgr5*) and fetal-like state (*Tacstd2* and *Sca1*) of WT organoids grown in glutamine-depleted media, in the presence or absence of ZAKα. *Hprt* is used as a reference. Mean and SEM are shown (*n* = 1 biological replicate accessed in technical triplicates). *p*-values were determined using a two-tailed *t*-test.

infected organoids which were cultured with either low dose Chx or in the absence of glutamine, and saw that in both cases ZAKα interacts with SRC (Fig. 4B & Supp. Fig. 9c). This interaction is decreased in the control organoids, highlighting its potential role in sensing ribosome collisions (Fig. 4C).

To understand the functional consequences of such interaction, we measured the levels of the active form of SRC by looking at the phosphorylation status of its residue Tyr416[53]. In all the conditions that caused a switch to a fetal signature, we saw a significant increase in SRC activity that was dependent on ZAKα (Fig. 4D). Furthermore, phosphorylation of YAP at a SRC specific site (Tyr357)[54] was increased in a SRC-dependent manner following ZAKα activation (Fig. 4E). Analysis of our RNA Seq data also revealed the presence of a YAP gene signature in *Rptor*-deficient organoids, compared with wild type cells (Fig. 4F), providing further evidence of its activation.

As there is a significant overlap between genes included in the YAP geneset used for this analysis and in the geneset used to identify fetal-like cells, we analyzed YAP-target genes that were not part of the fetal geneset, *Myof* and *Ankrd1*. Both genes were significantly upregulated upon *Rptor* deletion and this increase is mediated by ZAKα, supporting the idea that YAP/TAZ is activated upon ribosome impairment (Supp. Fig. 9d). In order to test whether the ZAKα-mediated activation of SRC and YAP caused the observed cell fate switch, we treated *Rptor*-deficient organoids with either a SRC inhibitor (dasatinib) or a YAP inhibitor (verteporfin). Both treatments could block the adult to fetal-like switch (Fig. 4G). Together, these results indicate that ribosome impairment causes a signaling cascade that results in the YAP-dependent activation of a fetal-like transcriptional program.

To assess whether ribosome-mediated activation of ZAKα caused the same cell identity switch in human tissue, we used human colonic

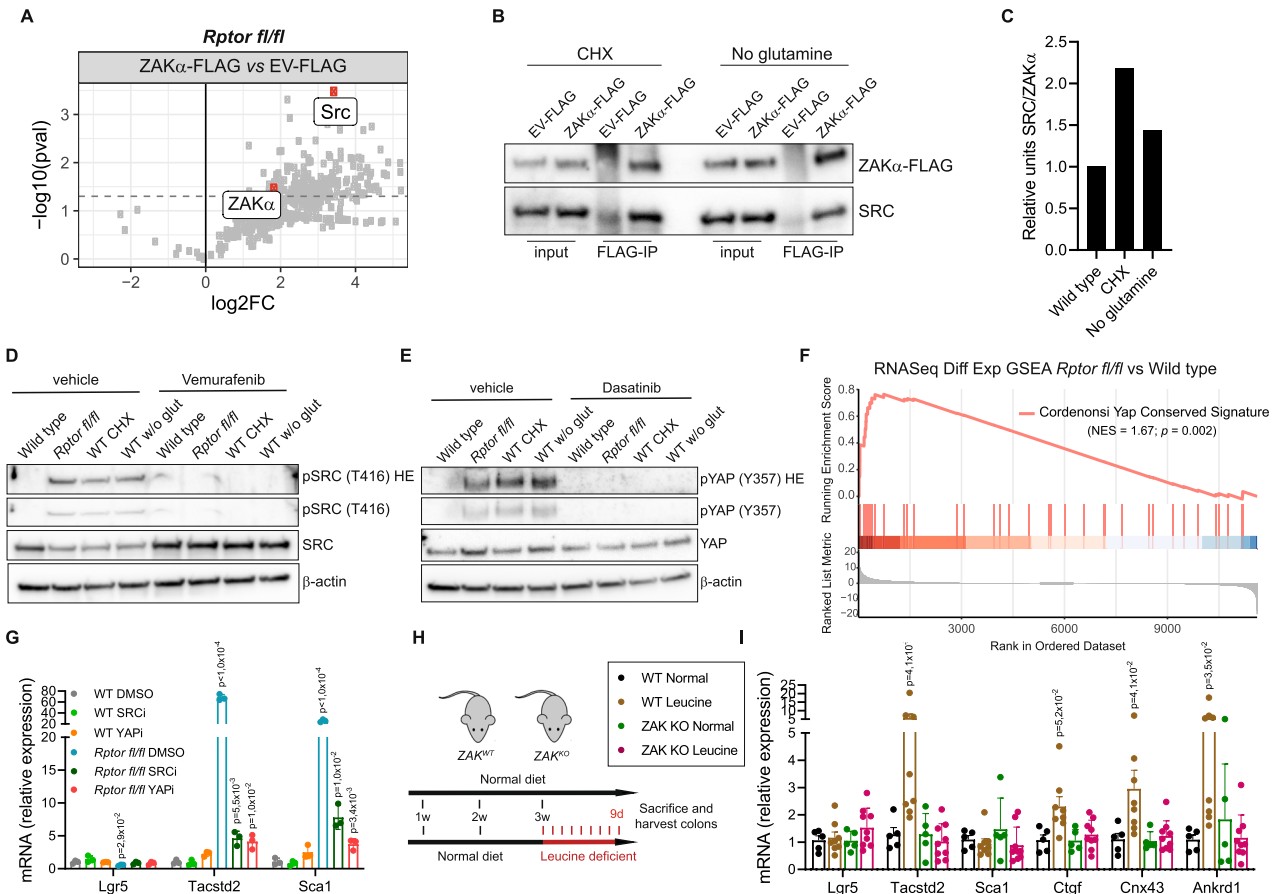

**Fig. 4 | ZAKα activates the SRC-YAP axis during ribosome impairment and results in ISCs identity switch in vivo during leucine deprivation. A** Volcano plot depicting differentially enriched interactors of ZAKα in *Rptor^{fl/fl}* organoids, using RIME-MS (*n* = 2 biological replicates). An empty vector tagged FLAG bait is used as a control. Significance when *p*-value ⩽ 0,05 (*t*-test, two-tailed). **B** Immunoprecipitation of ZAKα-FLAG confirms interaction with SRC in organoids treated with cyclohex-imide (0.015 ug/ml) and deprived from glutamine. EV_FLAG is used as a control. One biological animal was used. **C** Quantification of the western blot shows enrichment of ZAKα-SRC binding in organoids treated with cycloheximide (0.015 ug/ml) and deprived from glutamine, compared with wild type cells. **D** Western blot showing activation of SRC in *Rptor^{fl/fl}*, low dose cycloheximide and glutamine-deprived organoids compared to wild type. This activation is dependent on ZAKα activity, as inhibition with vemurafenib abolishes SRC phosphorylation (T416). β-actin serves as a loading control. Blue panel has a higher exposure. Blots were done on one animal. **E** Western blot showing activation of YAP in Rptor^{fl/fl}, low dose cycloheximide and glutamine-deprived organoids compared to wild type. This activation is dependent

on SRC, as inhibition with dasatinib abolishes YAP phosphorylation (Y357). β-actin serves as a loading control. Blue panel has a higher exposure. Blots were done on one animal. **F** Gene Set Enrichment Analysis based on RNASeq differential expression data comparing *Rptor^{fl/fl}* organoids to WT (*n* = 4 from 2 biological replicates for each). Enrichment is shown for transcriptional signatures related to YAP-target genes. *p*-values were determined using the *clusterProfiler* package. **G** RT-qPCR ana-lysis of genes related to stem (*Lgr5*) and fetal-like state (*Tacstd2* and *Sca1*) of WT and *Rptor^{fl/fl}*organoids treated with inhibitors for SRC (dasatinib, 100 nM, 24 h) and YAP (verteporfin, 3 uM for 24 h). *Hprt* is used as a reference. Mean and SEM are shown (*n* = 1 biological replicate accessed in technical triplicates). *p*-values were deter-mined using a two-tailed *t*-test. **H** Experimental workflow of the dietary interventions performed in WT and ZAK KO mice. **I** RT-qPCR analysis of genes related to stem (*Lgr5*) and fetal-like state (*Tacstd2* and *Sca1*) of colons of WT and ZAK KO mice. Hprt is used as a reference. Mean and SEM are shown (*n* = 5 (WT) and 9 (ZAK KO) bio-logical replicates each accessed in technical triplicates). *p*-values were determined using a two-tailed *t*-test.

organoids, which we grew in either full media or media deprived from leucine. In line with our previous findings, we saw that amino acid restriction led to an activation of the fetal signature which could be rescued when we inhibited ZAKα (Supp. Fig. 9e).

**Amino acid restriction in vivo drives the same phenotype switch**
Finally, as the intestine is an extremely complex tissue that cannot be fully recapitulated in organoid culture, we carried out an amino acid depletion study in WT and *Zak* knock-out mice (Fig. 4H & Supp. Fig. 9f). *Zak* deletion has no overt phenotype, and food intake is unchanged (Supp. Fig. 9g). We maintained animals on a leucine-deficient diet for 9 days (Fig. 4H), and measured the levels of *Lgr5* and markers of the fetal intestine (*Tacstd2, Sca1, Cnx43, Ctgf*) and YAP signature (*Ankrd1*) in the colons of these animals[55]. As observed in organoids, in vivo amino acid depletion resulted in an increase in fetal markers (Fig. 4I). Crucially however, deletion of *Zak* prevented the appearance of these cells, demonstrating that the change in cell fate caused by leucine

starvation is a ribosome impairment-mediated effect (Fig. 4I). It's worth noting that both in the human (Supp Fig. 8d) and mouse (Fig. 4E) colon samples, leucine depletion does not seem to affect the *Lgr5* + population. This may be due to the fact that this cellular population is rarer in the colon compared to the small intestine, and the effect of restricting amino acids may be underestimated as a result.

## Discussion
Ribosome stalls and collisions have long been known to cause mRNA and nascent protein degradation[56,57]. As they can be caused by defec-tive mRNA, it is clearly advantageous to a cell to remove the problem. The development of disome seq showed that collisions were far more common than expected, and confirmed their importance in the cor-rect folding of peptides, underlining the fact that they play a role in normal biology, alongside their role in sensing faulty mRNA[28,29]. Recent work has taken this a step further, and has shown that ribosome col-lisions may act as key signaling hubs that can determine cellular fate

by recruiting different players which activate specific pathways depending on the "severity" of the damage[41,45]. Indeed, studies have now started to dissect the downstream effects of ribosome impairment, and it appears that several factors, such as elongation rate, "strength" of stall, and the activity of the ribosome quality control mechanisms can subtly alter the signaling that results[44]. However, the extent of the in vivo relevance of ribosome impairment is currently unknown.

Here we show that ribosome-induced signaling through ZAKα is a crucial determinant of ISCs identity (Supp. Fig. 9h). Unlike canonical ZAKα signaling however, it appears to be independent of both JNK and p38 signaling, and instead it seems to be mediated by a SRC-YAP axis. Indeed, it has been shown that p38 activation is important for maintaining adult ISC identity[35], so it is not surprising that ZAKα activation of p38 does not mediate a switch to a fetal-like phenotype. Furthermore, contrary to what has previously been reported[41], we find that GCN2 and the ISR are not activated down-stream of ZAKα. However, Vind et al. have also found that GCN2 is not involved in ZAKα-mediated signaling[43], and a recent study has shown in yeast that GCN2-mediated ribosome quality control (RQC) is more likely to result from high levels of mRNA damage compared to low levels, demonstrating a coordination (and indeed and antagonism) between the RQC and ISR[58]. Crucially, activation of the ISR via ER stress is known to cause the differentiation of ISC[59], and thus would lead to the loss of the stem cell pool if it were activated by ribosome collisions.

Our results suggest an alternative mechanism through which intestinal stem cells can sense perturbations in ribosome dynamics that prompt them to switch to a fetal-like state. We describe a pathway activated by ribosome impairment, whereby ZAKα phosphorylates SRC, triggering a YAP/TAZ transcriptional reprogramming which converts adult ISCs into a fetal-like state. Several studies have implicated both YAP and TAZ in the maintenance of cell identity during different stress conditions[60]. The switch to a fetal-like state has also been observed in several stress conditions[10,11,14]. Yui et al. demonstrated that upon tissue damage, cells convert into a fetal-like state as a result of increased FAK/SRC signaling and consequent YAP/TAZ activation[11]. However, it is still not clear which specific damages and signaling cascades, lead to this transcriptional reprogramming. In this study, we propose that ribosome collisions may act as sensors for different stress conditions, allowing for the emergence of a fetal-like population of ISCs through the activation of ZAKα.

Further studies are clearly needed to understand the role of ribosomes and ZAKα in the ISC. The ISC is known to be a plastic population, and this plasticity is especially important under various stress conditions, when the cells can take on a new identity in order to overcome this[6]. A number of these stressors are known to result in the inhibition of translation, particularly calorie and nutrient deprivation[61,62]. Indeed, it has been shown that amino acid levels can have a significant effect on ISCs, although studies have focused on amino acid supplementation rather than depletion, and have been very limited in scope[63,64].

Here, we show that both deprivation of leucine or glutamine in vitro and leucine in vivo are sufficient to drive a ZAKα-mediated ISC identity switch, suggesting that ribosome dynamics may be used as a nutrient sensor to alter stem cell fate in response to local nutrient availability. As dedicated nutrient sensing machinery exists in the cell[65], it is surprising that the ribosome would be used in this way in the ISC. However, as mentioned above, activation of the ISR is detrimental to ISCs, and must be avoided if the stem cell population is to be maintained[59]. It is possible therefore that the recognition of amino acid depletion via changes in ribosome dynamics evolved in order to allow the cell to respond to such depletion, without activating the ISR, thus maintaining the vital ISC population. Although it has been shown that *Gcn2* deletion enhances intestinal inflammatory phenotypes, the specific role of GCN2 in the ISC was not analyzed in that manuscript[66],

however it is safe to say that the response to amino acid depletion in the intestine is a complex one. Interestingly, it has been shown that calorie restriction (as opposed to specific amino acid depletion) leads to an increase in the number of ISCs, rather than a decrease[24,67]. This process appears to be regulated in a non-autonomous manner, with Paneth cells working as direct sensors or calorie availability in this context, and activating pathways to augment ISC number. In our model, ribosome impairment via amino acid restriction, cycloheximide treatment, or inactivation of mTORC1 leads to a striking decrease of Paneth cells, perhaps explaining the difference between our findings and those related to calorie restriction.

The ability of ISCs to detect and respond to amino acid depletion is crucial to the well-being of the organ, and the sensor of this would be central to intestinal biology. Indeed, the fetal-like ISC phenotype has been shown to be important following helminth infection[10], extracellular matrix remodeling[11,12], and has been suggested to allow Wnt-independent growth in intestinal cancer[34,68]. We do not know whether ribosome collisions and ZAKα play a role in these models, but due to the diversity of triggers of this phenotype, it is possible that independent mechanisms can result in the same outcome.

In all, we show that ribosome impairment is a major result of translation inhibition in the intestine. This impairment results in a dramatic change in ISC identity both in vitro and in vivo, and we show that activation of a ZAKα-SRC-YAP axis is central to this. Considering the many stimuli that can drive such impairment, we propose that the dynamics of translation elongation represent a major cell fate decision checkpoint in the intestine, and a previously undescribed mechanism through which *Lgr5*-expressing cells can be targeted by therapy.

## Methods

### Ethical approval
All experiments were carried out with the approval of the relevant ethical bodies: NKI Animal Welfare Body, University of Copenhagen Institutional Animal Care and Use Committee, and the NKI-AVL Institutional Review Board (IRB).

### Mouse colonies
Animals for this study were bred in-house at the Netherlands Cancer Institute and all experimental protocols were approved by the NKI Animal Welfare Body.

C57BL/6 female and male mice between 8 and 12 weeks of age were used for experiments.

Animals were generated as previously described[69]. Briefly, for the recombination of *VillinCre*[ERT2]*Rptor*[fl/fl], *VillinCre*[ERT2]RPL22.HA and *VillinCre*[ERT2]*Rptor*[fl/fl]RPL22.HA animals, two consecutive injections of 80 mg/kg tamoxifen were performed and samples were taken after 120 h. For *Lgr5Cre*[ERT2]RPL22.HA and *Lgr5Cre*[ERT2]*Rptor*[fl/fl]RPL22.HA animals, a single intraperitoneal injection of 120 mg/kg tamoxifen and samples were taken after 24 h, in order to account for differences in recombination efficiency and total amount of cells.

For the leucine-deficient diet in vivo, WT and ZAK KO male mice around 10–11 weeks old were fed a full synthetic diet for 3 weeks. After 3 weeks, 5 mice (of each genotype) continued eating a full synthetic diet and 9 mice got a leucine-deficient diet for 9 days. Animals were sacrificed and the colons were extracted for gene expression analysis. This experiment was conducted at the Department of Experimental Medicine at the University of Copenhagen, and the protocol was approved by the Institutional Animal Care and Use Committee and performed in agreement with Danish and European regulations.

### Crypt culture
Small intestinal organoids were generated from isolated crypts collected from *VillinCre*[ERT2]*Rptor*[fl/fl], *VillinCre*[ERT2]RPL22.HA and *VillinCre*[ERT2]*Rptor*[fl/fl] RPL22.HA animals, as previously described[20]. Organoids were cultured in 30 μl BME (Amsbio) and grown in ENR media,

consisting of Advanced DMEM/F12 media (ThermoFisher Scientific) supplemented with 0.1% BSA (Sigma Aldrich), 1× N2 (ThermoFisher Scientific) and 1× B27 (ThermoFisher Scientific), together with 10% (v/v) Noggin and 10% (v/v) R-spondin. In order to have paired-samples to study the consequences of losing Raptor, an extra set of *Villin-Cre*[ERT2]*Rptor*[fl/fl] animals were induced in vitro.

To generate stem cell enriched (SCe) cultures, organoids were grown in ENR media supplemented with 10 μM of CHIR 99021 (Cayman Chemical) and 1.5 mM of valproic acid[33].

For the human organoids, a piece of normal colon tissue, of which the muscle layer and fat were removed, was cut in small pieces followed by EDTA treatment to release epithelium from underlying mesenchyme. The isolated epithelium was embedded in BME Cultrex and grown at 37 °C with ENR media, including Wnt conditioned media, until the organoids were fully formed. All patients gave written informed consent to have organoids generated from their left-over tissue, and agreed to have these included in a biobank for future scientific research. Human material cannot be shared due for ethical and privacy reasons. Protocols were approved by the NKI-AVL Institutional Review Board (IRB).

Cycloheximide (Sigma Aldrich) treatment was done using high (100 μg/ml) and low (0.015 μg/ml) doses for 30 min. GCN2 was inhibited using A-92 (2 μM for 24 h), p38 was inhibited with SB203580 (10 μM, 24 h) and JNK was inhibited with JNK-IN-8 (1 μM for 24 h). Dasatinib was used to inhibit SRC(100 nM, 24 h) and verteporfin was used to inhibit YAP (3 μM, 24 h). Finally, for inhibition of ZAK, Vemurafenib was added to the media at a final concentration of 1 μM and organoids were treated for 1 h.

## Immunofluorescence

Immunofluorescence of intestinal organoids was performed as previously described[70]. DAPI was combined with appropriate Alexa Fluor labeled secondary antibodies: Chicken anti-Rabbit Alexa Fluor 647 (ThermoFisher Scientific #A21443) 1:500; Donkey Anti-Rat Alexa Fluor 555 (Abcam #ab150154) 1:500; Donkey anti-Goat Alexa Fluor Plus 555 (ThermoFisher Scientific #A32816) 1:500. Images were collected on an inverted Leica TCS SP8 confocal microscope (Mannheim, Germany) in 12-bit with 25Xwater immersion objective (HC FLUOTAR L N.A. 0.95 W VISIR 0.17 FWD 2.4 mm). Imaris software (version 9.3.1, Oxford Instruments) was used for 3D reconstruction of images. The organoids were incubated with the following primary antibodies overnight: anti-Ly-6A (Sca1) (Biolegend #108101) 1:200, anti-Olfm4 (Cell Signaling #39141) 1:100, anti-Aldolase (Abcam #ab75751) 1:300, anti-Lysozyme (Agilent #A0099) 1:400, and anti-Mouse TROP-2 Antibody (R&D systems #AF1122) 1:50. Alexa Fluor labeled secondary antibodies (ThermoFisher Scientific #A22287) were combined with DAPI.

## Generation of intestinal organoids with ZAKα knock down, EV-FLAG and ZAKα-FLAG

shRNA targeting Zakα was chosen from the Open Biosystems Expression Arrest™ TRC library (target sequence CCACGATTATCTGAACCTGTT). As negative control an shRNA containing a scramble sequence was used (CAACAAGATGAAGAGCACCAA). These shRNAs were inserted into a pLKO.1 vector (Addgene).

The viral vectors were transfected into HEK293T cells (obtained from the ATCC) with third-generation packaging plasmids (pVSV-G, pRSV-REV and pMDL RRE). The viral supernatant was filtered through a 0,45-μm filter and concentrated using LentiX Concentrator (Takara).

For the affinity purification experiments, lentiviral vectors expressing FLAG, pLV[Exp]-Bsd-mPGK > {3xFLAG/Stuffer_300bp} (Ev-FLAG; ID = VB210329-1386htq) and pLV[Exp]-Bsd-mPGK > mMap3k20[NM_023057.5]/3xFLAG (ZAKα-FLAG; ID = VB210329-1381jwy), were constructed and packaged by VectorBuilder. Detailed information can be retrieved on vectorbuilder.com using the identifications.

Organoids were cultured in complete medium with growth factors and stem cell-inducing factors: 10 μM Rho kinase inhibitor Y-27632 (Cayman), 1 mM VPA (Biovision), 1 μM Jagged-1 (AnaSpec) and 6 μM CHIR99021 (Cayman). After 2 days in culture, organoids were then dissociated into single cells with TrypLE Express enzyme (Thermo Fisher Scientific). For the Raptor knock out organoids StemPro Accutase (Thermo Fisher Scientific) was used instead. Cells were resuspended in complete medium, containing growth factors, stem-cell inducing factors and 8 μg/ml polybrene and laid over wells covered with BME. The virus was added and cells were incubated in normal culture conditions. After 24 h of infection a layer of BME was put on top and the media was refreshed. The next day selection with 2 μg/ml puromycin started. Once selection was complete, the organoids were put in BME plugs and kept in normal culture conditions.

## Protein synthesis analysis

Organoids were grown in normal ENR media and taken at Day 4 for analysis. Protein synthesis rates were measured, as described previously[20]. Briefy, cells were treated with DMEM methionine-free media (ThermoFisher Scientific #21013024) for 20 min and incubated with 30 μCi/ml [35]S-methionine label (Hartmann Analytic) for 1 h. After washing the samples with PBS, proteins were extracted with lysis buffer (50 mM TrisHCl pH 7.5, 150 mM NaCl, 1% Tween-20, 0.5% NP-40, 1× protease inhibitor cocktail (Roche) and phosphatase inhibitor cocktail (Sigma Aldrich)) and precipitated onto filter paper (Whatmann) with 25% trichloroacetic acid and washed twice with 70% ethanol and twice with acetone. Scintillation was then read using a liquid scintillation counter (Perkin Elmer) and the activity was normalized by total protein content. All experiments were done in technical triplicates for each biological unit.

## RNA isolation and RT-qPCR

Organoids were washed twice with cold PBS and pellets were resuspended in TRIzol™ Reagent (ThermoFisher Scientific). RNA was isolated by chloroform extraction followed by centrifugation, isopropanol and glycogen precipitation and 75% ethanol washing. Pellets were resuspended in nuclease-free water and RNA was quantified with Nanodrop. Reverse transcription was performed using High-Capacity cDNA Reverse Transcription Kit (ThermoFisher Scientific) following manufacturer's instructions. qPCR was performed with SYBR™ Green PCR Master Mix (ThermoFisher Scientific) in technical triplicates and relative gene expression was calculated using the comparative CT method by normalization to *Hprt*. The primer sequences used were: *Hprt* forward (5′-CTGGTGAAAGGACCTCTCG-3′), *Hprt* reverse (5′-TGAAGTACTCATTATAGTCAAGGGCA-3′); *Lgr5* forward (5′-ACCCGCCAGTCTCCTACATC-3′), *Lgr5* reverse (5′-GCATCTAGGCGCAGGGATTG-3′); *Mex3a* forward (5′-ACACCACGGAGTGCGTTC-3′), *Mex3a* reverse (5′-GTTGGTTTTGGCCCTCAGA-3′); *Axin2* forward (5′-GCGACGCACTGACCGACGAT-3′), *Axin2* reverse (5′-GCAGGCGGTGGGTTCTCGGA-3′); *Lyz1* forward (5′-CGGTTTTGACATTGTGTTCGC-3′), *Lyz1* reverse (5′-GAGACCGAAGCACCGACTATG-3′); *Muc2* forward (5′-TGCCCAGAGAGTTTGGAGAG-3′), *Muc2* reverse (5′-CCTCACATGTGGTCTGGTTG-3′); *Alpi* forward (5′-ATGATGCCAACCGAAACCCC-3′), *Alpi* reverse (5′-GCGTGTCCTTCTCATTGGTAA-3′); *Tacstd2* forward (5′-GCTGATGCCGCCTACTACTT-3′), *Tacstd2* reverse (5′-CTACCGCTACCGAGACGACA-3′); *Sca1* forward (5′-CGAGGGAGGGAGCTGTGAGGTT-3′), *Sca1* reverse (5′-GAGGGCAGATGGGTAAGCAAAGAT-3′); *Spp1* forward (5′-CTCCATCGTCATCATCATCG-3′), *Spp1* reverse (5′-TGCACCCAGATCCTATAGCC-3′); *Cnx43* forward (5′-GAACACGGCAAGGTGAAGAT-3′), *Cnx43* reverse (GAGCGAGAGACACCAAGGAC-3′); *Myof* forward (5′-TGCTCATCCTGTTGCTGTTC-3′), *Myof* reverse (5′-GTTCTTCATTGCTTGCGTGA-3′); *Zakα* forward (5′-GAGAAGTGGATCGTGGGAATAG-3′), *Zakα* reverse (5′-TTTGGGAACGCTGT

AGAGAAG-3'); *Zakβ* forward (5'-GGCCATCGTTCAAGCAAATC-3'), *Zakβ* reverse (5'-GATGAGGGAAACGCCCTAAA-3').

## RNA sequencing

Total RNA was isolated from wild type, *Rptor^fl/fl* and SC-enriched organoids as described above. Quality of the samples was measured with the 2100 Bioanalyzer using a RNA Nanochip (Agilent) and accepted for downstream analysis when showing an RNA integrity number (RIN) above 8. Libraries were generated with the TrueSeq Stranded mRNA kit (Ilumina) and sequenced using the HiSeq2500 equipment.

## Affinity purification and quantitative mass spectrometry

Rapid Immunoprecipitation mass spectrometry of endogenous proteins (RIME) was used to identify ZAKα interactors as previously described[71], with some minor modifications. Briefly, wild type and Rptor^fl/fl organoids transfected with EV-FLAG and ZAKα-FLAG constructs were crosslinked with 1% formaldehyde for 10 min and quenched with glycine to stop the reaction. Cells were lysed with LB3 and left in rotation for 10 min at 4 °C. Lysates were sonicated for 10 cycles of 30 s ON/30 s OFF and cleared by centrifugation at max speed for 12 min, at 4 °C. Samples were incubated with pre-washed FLAG-conjugated beads, overnight at 4 °C. For mass spectrometry, peptide mixtures were prepared and measured as previously described[72], with the following exceptions. Peptide mixtures (10% of total digest) were loaded directly onto the analytical column and analyzed by nanoLC-MS/MS on an Orbitrap Fusion Tribrid mass spectrometer equipped with a Proxeon nLC1200 system (Thermo Scientific). Solvent A was 0.1% formic acid/water and solvent B was 0.1% formic acid/80% acetonitrile. Peptides were eluted from the analytical column at a constant flow of 250 nl/min in a 104 min gradient, containing a 84-min stepped increase from 2% to 24% solvent B, followed by an 20 min wash at 80% solvent B.

Raw data were analyzed by MaxQuant (version 2.0.3.0)[73] using standard settings for label-free quantitation (LFQ). MS/MS data were searched against the Swissprot Mus musculus database (17,073 entries, release 2021_04) complemented with a list of common contaminants and concatenated with the reversed version of all sequences. The maximum allowed mass tolerance was 4.5ppm in the main search and 0.5 Da for fragment ion masses. False discovery rates for peptide and protein identification were set to 1%. Trypsin/P was chosen as cleavage specificity allowing two missed cleavages. Carbamidomethylation was set as a fixed modification, while oxidation and deamidation were used as variable modifications. LFQ intensities were Log2-transformed in Perseus (version 1.6.15.0)[74], after which proteins were filtered for 2 out of 2 valid values in at least one sample group. Resulting data was post-processed with a custom R script for quantile normalization, data imputation and comparisons across conditions.

## Western Blot analysis

Organoids were washed twice with cold PBS and pellets were resuspended in lysis buffer (Urea 6 M, SDS 1%, NaCl 125 mM, 25 mM TrisHcl pH = 8, 1× protease inhibitor cocktail (Roche) and phosphatase inhibitor cocktail (Sigma Aldrich)) on ice and sonicated for 10 cycles of 1 s ON/1 s OFF with an amplitude between 20% to 30%. Proteins were quantified using the Pierce™ BCA Protein Assay Kit (ThermoFisher Scientific), separated by SDS-PAGE and transferred to 0.2 μm pore nitrocellulose membranes (PALL). Membranes were blocked with 5% milk/TBST for 45 min at room temperature and incubated with primary antibodies overnight at 4 °C followed by the appropriate secondary antibodies conjugated to HRP for 1 h at room temperature. Proteins were visualized by ECL-Plus (ThermoFisher) and Syngene. The antibodies used in this study were: Raptor (Cell Signaling Technology #2280) 1:1000, bActin (Sigma Aldrich #A2228) 1:1000, pS6 (Ser235/236) (Cell Signaling Technology #4858) 1:1000, HA (Cell Signaling

Technology #3724) 1:1000, RPL22 (Novus Biologicals #NBP1-98446) 1:1000, RPL7 (ThermoFisher Scientific #PA5-36571) 1:1000, ZAK (Proteintech #14945-1-AP) 1:500, gTubulin (Cell Signaling Technology #5886) 1:1000, p150 (BD Biosciences #610473) 1:1000, peIF2a (Ser51) (Cell Signaling Technology #9721) 1:1000, SRC (Cell Signaling Technology #2108) 1:1000, pSRC (Tyr416) (Cell Signaling Technology #2101) 1:1000, YAP (Santa Cruz Biotechnology #sc-15407) 1:1000, pYAP (Y357) (Abcam #ab254343) 1:1000, and FLAG (Sigma #F3165) 1:1000.

## Bioenergetics Analysis

Oxygen consumption rates (OCR) and extracellular acidification rates (ECAR) were measured using Seahorse Bioscience XFe24 Analyzer (Agilent). Organoids were first seeded in normal plates, in order to diminish growth variations. At Day 3, organoids were resuspended in 40ul of BME, re-plated in XF24 Seahorse Cell Culture microplates (Agilent) and cultured for 16–24 h before the analysis, which was done according to the manufacturer's instructions in DMEM (Sigma-Aldrich) supplemented with 2 mM L-glutamine for the ECAR experiments and additional 5.5 mM D-glucose for the OCR measurements. For the ECAR analysis, the following reagents were added: glucose (10 mM), oligomycin (1 μm) and 2-deoxy-D-glucose (50 mM). For OCR measurements, the following reagents were added: oligomycin (1 μM), FCCP (0.4 μM), and rotenone (1 μM) and antimycin A (1 μM). When applicable, cells were pre-treated with either vemurafenib (1 μM) or cycloheximide (0.015 μg/μl) for the desired time. All results were normalized to DNA content. Briefly, cells were scraped from the microplates and centrifuged for 3 min at 4,000 rpm at 4 °C. Pellets were then washed with cold PBS and resuspended in lysis buffer (75 mM NaCl, 50 mM EDTA, 0.02% SDS, 0.4 mg/ml Proteinase K). Samples were incubated at 56 °C for 2 h. Isopropanol (1 volume) was then added and mixed and incubated at 4 °C overnight. Tubes were centrifuged at 8,000 rpm for 30 min at 4 °C and pellets washed with 70% ethanol and air dried. DNA was resuspended in $H_2O$ and quantified using Nanodrop.

## Ribosome profiling

**Sample preparation.** Samples were prepared as described previously[69]. Briefly, for in vitro analyses, organoids were generated from *VillinCre*^ERT2^RPL22.HA and *VillinCre*^ERT2^*Rptor^fl/fl*RPL22.HA animals. Around 120–150 plugs of 30 μl BME were used for each replicate. Cells were treated with 100 μg/ml cycloheximide for 3–5 min at 37 °C and immediately incubated on ice for the remainder of the experiment. After collecting the cells, pellets were washed twice with cold PBS supplemented with 100 μg/mL cycloheximide, resuspended in ice-cold lysis buffer (20 mM Tris Hcl pH 7.4, 10 mM MgCl2, 150 mM KCl, 1% NP-40, 100 μg/mL cycloheximide and 1x EDTA-free proteinase inhibitor cocktail (Roche)) and incubated for 20 min on ice. Lysates were then centrifuged at max speed for 20 min at 4 °C and the supernatants were collected.

For in vivo analyses, *Lgr5Cre*^ERT2^RPL22.HA and *Lgr5Cre*^ERT2^*Rptor^fl/fl* RPL22.HA animals were euthanized by $CO_2$ and small intestines were immediately dissected, flushed with cold PBS supplemented with 100 μg/mL of cycloheximide and snap frozen using liquid nitrogen. Frozen samples were then homogenized by pestle and mortar while submerged in liquid nitrogen and the resulting powder was resuspended in ice-cold lysis buffer and incubated on ice for 30 min. Finally, lysates were centrifuged at max speed for 20 min at 4 °C and the supernatants were collected.

**HA pulldown.** All lysates were incubated with Pierce™ Control Agarose Matrix (ThermoFisher) for 20 min at 4 °C in order to remove any non-specific binding, following by an incubation with pre-washed AntiHA.11 Epitope Tag Affinity Matrix (BioLegend) for 4 hours (in vitro samples) or overnight (in vivo samples) at 4'C. Tagged ribosomes were then

eluted from the beads by incubating with 200 μg/mL HA peptide (ThermoFisher Scientifc) for 15 min at 30'C with constant agitation. Non-protected RNA was digested with 10 ul of RNase I (ThermoFisher Scientific) for 40 min at 25 °C and the reaction was stopped by adding 13 ul of SUPERASE (ThermoFisher Scientific). Ribosome protected fragments (RPFs) were finally purified using miRNeasy minikit (Qiagen) accoridng to manufacturer's instructions.

**Library preparation.** The library preparation was performed as previously described[75] with some modifications. To discard undigested RNA fragments, RPFs were run in a 10% TBE-Urea polyacrylamide gel and size selected between 19 nt and 32 nt using marker RNAs. 3' ends were dephosphorylated using T4 polynucleotide kinase (PNK) (NEB) and 1.5xMES buffer (150 mM MES-NaOH, 15 mM MgCl2, 15 mM β-mercaptoethanol and 450 mM NaCl, pH 5.5) and incubated at 37 °C for 4 h. After purifying RNAs with Trizol, 3'adpaters were ligated with T4 RNA ligase I (NEB) and incubated overnight at 24'C. After size-selecting the ligated products, 5' ends were phosporylated with T4 PNK and incubated at 37'C during 30 min. 5' adapters were then ligated with T4 RNA ligase I and incubated 37'C for 2.5 h. The RPFs containing both adapters were then size-selected one more time and rRNA depletion was performed by first denaturing the samples at 100 °C for 1 min followed by incubation with 10uM biotinylated oligos (5'-UGAUCU-GAUAAAUGCACGCAUCCCCCC-3'; 5'-CGUGCGAUCGGCCCGAGGUUA UCUAGAGUCACCA-3'; 5'-AUUCCAUUAUUCCUAGCUGCGGUAUCCAG GCGGCUC-3'; 5'-GGGCCUCGAUCAGAAGGACUUGGGCCCCCCACGA-3'; 5'-UGGCUUCCUCGGCCCCGGGAUUCGGCGAAAGC-3'; 5'-ACG-GACGCUUGGCGCCAGAAGCGAGAGCCCCUCGGG-3'; 5'-ACCCGGCUA UCCGGGGCCAACCGAGGCUCCUUCGCG-3'; 5'-ACCGACGCUCAGA-CAGGCGUAGCCCCGGGAGGA-3'; 5'-GGCGGACGGGGGGAGAGGGA-GAGCGC-3'; 5'-GGCGAGACGGGCCGGUGGUGCGCCCUCGGC-3'; 5'-CCAGAAGCAGGUCGUCUACGAAUGGUUUAG-3'; 5'-AUCCCCGAUCCC CAUCACGAAUGGGGUUCA-3'; *for the* in vivo *samples also added the following*: 5'-GCUCUGCUACGUACGAAACCCCGACC-3'; 5'-GUGUCGAG GGCUGACUUUCAAUAGAUCG-3'; 5'-AGAUCCAACUACGAGCUUUUUA ACUG-3'; 5'-AACGCCACUUGUCCCUCUAAGAAGU-3'; 5'-CAAGUGCGU UCGAAGUGUCGAUGAUC-3') in 2xSSC buffer (ThermoFisher Scientific) for 15 min at 37 °C. Pre-washed MyOne Streptavidin C1 DynaBeads (ThermoFisher Scientific) were then incubated with samples in wash/bind buffer (2 M NaCl, 1 mM EDTA, 5 mM Tris and 0.2% Triton X-100) for 30 min at 37 °C with agitation. Supernatants were collected and RPFs were purified and re-suspended in 8 μL of RNase-free water. cDNA was synthesized with SuperScript III (ThermoFisher Scientific) according to manufacturer's instructions and using the RTP primer (seq). After purification with G50 columns (Merck), cDNA was amplified using Phusion High-Fidelity DNA Polymerase (ThermoFisher Scientific) for 18 cycles, with primers containing different indexes to allow for sequencing[69]. PCR products were purified using the QIAquick PCR purification kit (Qiagen) followed by a final size selection with a E-Gel SizeSelect II 2%, (ThermoFisher Scientific). All samples were accessed with the Agilent 2100 Bioanalyzer to assure high quality and quantify the molarity and the libraries were sequenced on the Illumina HiSeq2500.

**Data analysis.** Initial quality control of all sequencing data was performed using the *FastQC* tool. Adapter trimming and size selection marker cleaning of the raw Ribo-seq data was performed using the *cutadapt* tool[76]. After Ribo-seq reads were cleaned from rRNA fragments, QC plots were generated with *RiboCode*[77] for which Ribo-seq reads were mapped to mm10 genome using the *STAR* aligner[78]. For the quantification of transcript abundances with RNAseq, *Salmon*[79] was used with protein-coding transcript sequences obtained from gencode vM21 annotation. To measure ribosome occupancy with Ribo-seq, the same tool was used with a different reference sequence set, where UTRs were trimmed from the same mRNA sequences and duplicated

sequences were removed prior to the run. Differential expression and ribosomal occupation analyses were performed in R environment, using the *DESEQ2* package[80] with Salmon outputs. Differential translation efficiency analysis was performed with RiboDiff[81] for which the input consisted of Ribo-seq and RNAseq transcript quantifications (NumReads column in salmon output) of primary transcripts, excluding the genes with low sequencing depth. Primary transcripts were decided based on Ensembl 96 APPRIS annotation. Gene Set Enrichment Analyses were performed in R using the *clusterProfiler* package[82] for which the genesets were taken from MSigDB v7.0 with additional genesets from published sources[11]. For the differential codon usage analysis performed with Ribo-seq data, we use the assumption that RPFs' P-site is at the nucleotide position 12, 13 & 14 for all in-frame reads, with a correction of +1 or −1 for out-of-frame reads.

## Reporting summary
Further information on research design is available in the Nature Research Reporting Summary linked to this article.

## Data availability
The sequencing datasets generated and analysed during the current study are available in the NCBI Gene Expression Omnibus repository, and can be accessed using the GSE180208 accession id. The proteomic datasets are available in the PRIDE repository, and can be accessed using the PXD033122 accession id. Source data are provided with this paper.

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

## Acknowledgements

We thank Drs. Vivian S. W. Li and Pedro Antas (both Crick Institute, United Kingdom) for generating and sharing the ZAK KO mouse, and Dr. Christoffer Clemmensen and Charlotte Svendsen (both University of Copenhagen, Denmark) for help with dietary interventions in mice. Work in the Faller lab is supported by the KWF (NKI-2016-10535,NKI-2021-13878), and the NWO (OCENW.KLEIN.263). JS is supported by an EMBO Long Term Fellowship [210-2018]. Work in the Bekker-Jensen lab was supported by the European Research Council (ERC) under the European Union's Horizon 2020 research and innovation program (grant agreement 863911 - PHYRIST). S.J.E.S. and A.K.G. were financially supported by the K.W.F. (Young Investigator Grant 11491/2018-1). We would like to acknowledge the NKI- AVL Core Facility Molecular Pathology & Biobanking (CFMPB) for supplying NKI-AVL Biobank material and lab support.

## Author contributions

J.S. and W.J.F. conceived the study and wrote the manuscript. F.A. carried out all the bioinformatic analysis. J.S., S.R., G.S., S.P., A.K.G., S.H.-P., R.v.d.K., D.B., L.H., and S.J.E.S. carried out experiments. W.J.F., M.A., W.Z., and S.B.J. supervised experiments.

## Competing interests

The authors declare no competing interests.
