## [Peer Review File · Nature Communications]

Ribosome impairment regulates intestinal stem cell identity via ZAK α activationEditorial Note: Parts of this Peer Review File have been redacted as indicated to remove third-party material where no permission to publish could be obtained.

REVIEWER COMMENTS

Reviewer #1 (Remarks to the Author):

Silva et al generate a mouse model with disruption of the raptor gene in intestine induced by Vil/Cre and obtain organoids from the small intestine. Comparing these with wild type organoids they observe differences in the cellular content of the organoids except in the goblet and enterocytic Alpi+ cells. They observe loss of Lgr5+ stem cells and Paneth cells and gain of fetal-like stem cells. All this demonstrated by qRT-PCR and RNA-seq. Then they show the presence of Ribosome collisions in the raptor knockdown organoids and in vivo. They further demonstrate that this phenotype is phenocopied by low cycloheximide treatment and no glutamine or no leucine. This phenotype is mediated by Zaka signalling. The study is interesting and relevant reinforcing ribosome dynamics with nutrient sensing and ISC identity. Some issues could be improved:

- In Introduction authors should also reference the different stromal cell types that have been identified as sources of Wnt ligands production, besides Paneth cells.

- Previous works addressing fetal intestinal markers are also missing: Mustata et al. (2013), Fordham et al. (2013).

- Authors state that "It is currently unknown if ISCs are sensitive to changes in translation.". There are already evidences in the literature reporting phenotypic impact over ISCs by modulating different players involved in the translational machinery, namely the mTOR pathway and specific RNA-binding proteins: Madison et al. (2013).

Wang et al. (2015). Transformation of the intestinal epithelium by the MSI2 RNA-binding protein. *Nat Commun.* 6:6517. DOI: 10.1038/ncomms7517.

Yousefi et al. (2016). Msi RNA-binding proteins control reserve intestinal stem cell quiescence. *J Cell Biol.* 215(3):401-413. DOI: 10.1083/jcb.201604119.

Igarashi and Guarente (2016). mTORC1 and SIRT1 Cooperate to Foster Expansion of Gut Adult Stem Cells during Calorie Restriction. *Cell.* 166(2):436-450. DOI: 10.1016/j.cell.2016.05.044.

Pereira et al. (2020). MEX3A regulates Lgr5+ stem cell maintenance in the developing intestinal epithelium. *EMBO Rep.* 21(4):e48938. DOI: 10.15252/embr.201948938.

- In the results, in the first experiment, what is the impact of Raptor loss in other cell populations besides stem cells?

- The results in organoids are based on qRT-PCR. It would be nice to see in situ staining of some the stem adult and fetal markers, paneth cells. It would also be relevant to see which cells express Raptor in vivo and in organoids.

- In the Lgr5CreERT2 mouse strain used authors should demonstrate that 24h after tamoxifen induction the Lgr5+ stem cells population is still present in the crypts. In the knockdown model, the Lgr5+ is a transition cell?

- The western blot showing Zaka phosphorylation and Zaka expression could be improved. Zaka phosphorylation upon glutamine deprivation is not shown.

- Authors do not explain the rationale in addressing Zaka KO effect in the colon (Fig. 6) instead of the small intestine, used for all the previous experiments. Moreover, in this experimental setup interpretation must be cautious as no decrease in Lgr5 expression or an increase in Sca1 are observed upon leucine removal.

- In the discussion authors should reconcile their observations with previous works by the Sabatini and Yilmaz groups showing that calorie restriction mediated by reduced mTOR pathway activity leads to Lgr5+ ISCs increased number due to a bystander effect on Paneth cells?

- It is surprising that Zaka KO or knockdown per se does not lead to any phenotype. If ribosome stalling and collisions have an important physiological role in sensing faulty mRNAs, Zaka modulation should produce some type of cellular phenotype. Are there alternative pathways mediating this process? This should be discussed.

- Authors do not show how collision-induced signaling through Zaka specifically targets glycolysis-related genes in ISCs only.

Reviewer #2 (Remarks to the Author):

Silva et al. presented exciting and captivating observations in which Zaka-dependent, yet Gcn2-eIf2 α -independent, pathway results in ISCs to switch cell identity. Given that growing understanding of the ribosome collision as a sensor for cellular status, the finding of the involvement in intestinal stem cell identity is quite timely and will appeal to board readers. However, the discussion about ribosome profiling data and ribosome collision was not helpful in the authors' reasoning. In addition, the direct detection of ribosome collisions was lacking. The following concerns are needed to be addressed before publication.

Major comments:

1. There was no direct evidence linking ribosome collisions to the observed phenotype. Although authors speculated that all the conditions (Raptor depletion, low concentration of cycloheximide, and depletion of amino acids) may incur ribosome collisions, no direct evidence of ribosome collision was provided. It is necessary to show solid verification of ribosome collisions by, for example, sucrose density gradient followed by RNase digestion to probe the increase of RNase-resistant disomes. Sequencing of the disome footprints (i.e., disome profiling/disome-seq) could be an alternative approach.

2. Fig. 2G showed that footprint accumulation on the stop codon. First, this data was difficult to interpret. The authors should consider another way of data visualization. Second, the data suggested the accumulation of footprints that hold stop codons at P site, in addition those at A site. This seemed to be unusual since the ribosome should be dissociated from mRNA where the A site lies at the stop codon. Ribosomes on P-site stop codon may represent the post-termination, unrecycled ones. The author should carefully characterize the footprints.

3. Related to the point above, the ribosome pausing on the stop codon was not supported by the meta-gene analysis in Supplemental Fig 4C.

4. Not simply focusing on the stop codon, a more comprehensive survey of ribosome pauses induced by Raptor depletion should be carefully conducted (e.g., codon/amino acid features associated with the pausing).

5. Regarding Raptor depletion and inactivation of mTORC1, the most straightforward impact in translation would be reduced translation initiation, mediated by LARP1 and/or 4EBP1 (especially, dephosphorylated LARP1 would lead to the strong repression of ribosome protein translation.) However, the contribution of this effect was not discussed in the manuscript (regarding the phosphorylation status and translation efficiency changes in ribosome profiling/RNA-seq data). Authors may speculate the larger impact on elongation (such as reported in authors' earlier work of

Faller et al. Nature 2015) through eEF2 phosphorylation. However, this point was not discussed in the manuscript. Again, the effect on the termination/ribosome recycling as described in Fig. 2G was not linked to either initiation or elongation scenario. The authors were requested to provide a reasonable explanation for their observations on inefficient termination/recycling by Raptor depletion.

Minor comments:

1. The TE of glycolysis-related genes increased upon Raptor depletion. Yet, the author should provide evidence at the protein level. Western blot or mass spectrometry would be helpful to strengthen the authors' conclusion. The same also applied to oxidative phosphorylation-related genes.
2. In line 66, although this reviewer understood that the use of phrase "the first evidence" was intended to stress the importance of the findings, more objective phrases are recommended, since such a wording may not fit to the journal policy.
3. In line 123, the citation of paper 41 at the phrase of "Upon further analysis of this data⁴¹," was not clear whether this indicated the data analysis of earlier paper or that obtained in this study. Please clarify this.
4. The detail for the RNA-seq experiments was not described in the method section. This reviewer imagined that the experiments were conducted in whole organoid not in HA-tag isolated ribosome fraction as done in ribosome profiling, however, the details should be described.
5. Related to this, mRNA areas used for read counting for ribosome profiling and RNA-seq were not clear in the method. Given that ribosome footprint originates mainly from ORF, the read counting of RNA-seq should be also restricted to the ORF for fair comparison. Please clarify this point in the method. If the entire mRNA region was used for the read counting, the reanalysis of the data is recommended.

Reviewer #3 (Remarks to the Author):

General comments:

In this manuscript, Silva and colleagues examine the effect of translation inhibition on the identity of intestinal stem cells (ISCs). They make the interesting claim that by activating $Zak\alpha$, ribosome collisions drive a change in ISCs to a fetal-like identity. In the first initial experiments, the authors took advantage of organoids with a conditional knock out of Raptor to inhibit mTORC1. Upon the loss of Raptor, these cells reverted to a fetal-like identity and, more importantly, rewired their metabolism from oxidative phosphorylation to glycolysis. This switch appears to be regulated through translation and not transcription. The authors then went to show ribosome stalling through the addition of cycloheximide results in a similar phenotype and make the conclusion that it is likely to be collisions since addition of cycloheximide to high concentrations does not trigger this switch. This then led the authors to examine the role of $Zak\alpha$, which has recently been shown to be activated by ribosome collisions, in the process. The authors showed that the factor is important for the switch in the identity, as well as metabolic rewiring, but that its canonical function as a MAPKKK is dispensable for this activity.

These studies expand on the newly identified roles of ribosome collisions on triggering diverse signaling pathways, and as such, its subject is topical and relevant to a wide audience. However, the manuscript in its current form lacks important experiments and controls to support the bold conclusions made by the authors. Indeed, the authors make the argument that collisions are responsible for this switch using one experiment – they show that low concentrations of cycloheximide

also drive the switch. However, there was no effort to show that collisions took place under these conditions. Furthermore, the authors did not attempt to investigate the mechanism by which Zak α activation results in changes in translation efficiency.

Specific comments

There are several obvious experiments that the authors can do to bolster their arguments/conclusions:

- 1- The authors need to show that deletion of Raptor results in ribosome collisions; currently this is all based on conjecture. They also need to show that cycloheximide addition at the concentration used in their studies results in ribosome collisions.
- 2- How does the activation of Zak α affect ribosome function? In other words, the factor is a MAPKKK, but the authors argue that this function is unrelated to what they observe here. So how does it affect the ribosome's ability to translate certain genes, but not others?
- 3- The authors only used cycloheximide as a probe for ribosome collision. Why only cycloheximide, especially given that usage of the drug has many caveats and others in the ribosome-collision field use other drugs? To that end, amino acid starvation does not only trigger ribosome collisions, but it also inhibits mTORC1, so this becomes a circular argument, leading us back to the effects seen in Figures 1 and 2 in the absence of Raptor.
- 4- The authors need to convincingly show that GCN2 can be activated in their system using other stressors.
- 5- The phosgel in Figure 3C is not convincing. The same gels used by the Green group show a much greater shift for Zak α when phosphorylated.

REVIEWER COMMENTS

Reviewer #1 (Remarks to the Author):

Silva et al generate a mouse model with disruption of the raptor gene in intestine induced by Vil/Cre and obtain organoids from the small intestine. Comparing these with wild type organoids they observe differences in the cellular content of the organoids except in the goblet and enterocytic Alpi+ cells. They observe loss of Lgr5+ stem cells and Paneth cells and gain of fetal-like stem cells. All this demonstrated by qRT-PCR and RNA-seq. Then they show the presence of Ribosome collisions in the raptor knockdown organoids and in vivo. They further demonstrate that this phenotype is phenocopied by low cycloheximide treatment and no glutamine or no leucine. This phenotype is mediated by Zakalpha signalling. The study is interesting and relevant reinforcing ribosome dynamics with nutrient sensing and ISC identity. Some issues could be improved:

We thank the reviewer for their constructive comments.

- In Introduction authors should also reference the different stromal cell types that have been identified as sources of Wnt ligands production, besides Paneth cells.

We have updated the introduction to highlight the additional sources of Wnt ligands.

- Previous works addressing fetal intestinal markers are also missing: Mustata et al. (2013), Fordham et al. (2013).

Although the Mustata reference was included in the original manuscript, it was not included in the introduction section. We had failed to include the Fordham reference however. This has been corrected and both are now included.

- Authors state that "It is currently unknown if ISCs are sensitive to changes in translation.". There are already evidences in the literature reporting phenotypic impact over ISCs by modulating different players involved in the translational machinery, namely the mTOR pathway and specific RNA-binding proteins: Madison et al. (2013).

Wang et al. (2015). Transformation of the intestinal epithelium by the MSI2 RNA-binding protein. Nat Commun. 6:6517. DOI: 10.1038/ncomms7517.

Yousefi et al. (2016). Msi RNA-binding proteins control reserve intestinal stem cell quiescence. J Cell Biol. 215(3):401-413. DOI: 10.1083/jcb.201604119.

Igarashi and Guarente (2016). mTORC1 and SIRT1 Cooperate to Foster Expansion of Gut Adult Stem Cells during Calorie Restriction. *Cell*. 166(2):436-450. DOI: 10.1016/j.cell.2016.05.044.

Pereira et al. (2020). MEX3A regulates Lgr5+ stem cell maintenance in the developing intestinal epithelium. *EMBO Rep*. 21(4):e48938. DOI: 10.15252/embr.201948938.

The reviewer is correct in their assertion that several players known to be involved in mRNA translation have been associated with ISC dynamics. We would like to note though that while these factors have been studied, the link to translation is rarely probed in these publications. For example, several of the papers listed above refer to the role of mTOR signaling in ISCs. These publications do not go beyond the mTOR kinase, and do not assess translation. mTOR is known to regulate many processes independently of its regulation of translation (metabolism and autophagy, for example), so it is not shown in these publications that the effects of mTOR are mediated via translation.

The reviewer is correct that we should have highlighted these studies however, and we have updated the manuscript to do this.

- In the results, in the first experiment, what is the impact of Raptor loss in other cell populations besides stem cells?

Raptor loss in this experiment causes very significant changes in the cellular population of the organoids, and we have now included that data. We believe that the Raptor-deleted organoids are almost entirely composed of fetal-like stem cells, and we have now included qPCRs, RNASeq, and immuno-fluorescence to support this (Fig. 1E and Supplementary Fig. 1B&2B). In short, this shows that markers of adult stem cells, paneth cells, and enterocytes are all decreased following Raptor deletion, while markers of fetal-like stem cells, tuft cells and goblet cells are increased.

- The results in organoids are based on qRT-PCR. It would be nice to see in situ staining of some the stem adult and fetal markers, paneth cells. It would also be relevant to see which cells express Raptor in vivo and in organoids.

See response above. We have stained for Olfm4 (adult stem cells), Sca1, Tacstd2 (fetal-like stem cells), Lysozyme (Paneth cells), and AldolaseB (enterocytes), and have included this data in Fig. 1E and Supplementary Fig. 1B.

- In the Lgr5CreERT2 mouse strain used authors should demonstrate that 24h after tamoxifen induction the Lgr5+ stem cells population is still present in the crypts. In the knockdown model, the Lgr5+ is a transition cell?

This is indeed an important point, and highlights a major difference between the organoid and *in vivo* model. To test this, we made use of the fact that in this model, Lgr5 is linked to GFP, allowing the visualization of Lgr5-expressing cells. 24-hours after Raptor deletion in the intestine

(the same time point used for the RiboSeq), there are still Lgr5+ stem cells present, as can be seen in the image below.

RL Fig. 1: GFP staining shows that crypt base still contains Lgr5-expressing cells after Raptor deletion

- The western blot showing Zaka phosphorylation and Zaka expression could be improved. Zaka phosphorylation upon glutamine deprivation is not shown.

The Zaka blot is a very difficult one to do. We do not use phospho-gels (which are used by the Green lab), but we think that the shift in band size is clear, and is of a similar magnitude to that in other publications (PMID: 35316659, PMID: 32289254). Furthermore, we believe that the combination with the functional consequences of shZak show beyond doubt that Zaka is activated following ribosome impairment. We have added the FLAG tagged Zaka western below as a confirmation experiment (RL Fig. 2)

Additionally, we have now added the same western for the Glutamine and Leucine deprivation conditions (Fig. 3C).

RL Fig. 2: Western blot using a FLAG antibody showing that Cycloheximide treatment, glutamine deprivation, and Raptor deletion all cause a shift in the band, indicative of phosphorylation.

- Authors do not explain the rationale in addressing Zaka KO effect in the colon (Fig. 6) instead of the small intestine, used for all the previous experiments. Moreover, in this experimental setup interpretation must be cautious as no decrease in Lgr5 expression or an increase in Sca1 are observed upon leucine removal.

The reviewer makes a good point here. The decision to focus on the colon in this case was driven by two arguments:

First, while we have shown that the fetal signature is present *in vivo* in the small intestine following Raptor deletion (Fig. 2F), the magnitude of increase is less than that in the organoids, probably due to the retention of the Lgr5+ population, as shown in a response above. Alongside this, it is known that the fetal signature is easier to induce in the colon (PMID: 34103493; private communication from several labs; supporting data below in RL Fig. 3), suggesting that this tissue is probably the best to look at in more complex situations.

Second, the human organoids we used were also of colonic origin, and we wanted to maintain consistency in this “*in vivo* relevance” part of the manuscript.

We have updated the text to ensure that this is clear in the manuscript.

[redacted]

RL Fig 3: Relative expression of fetal markers in organoids from the colon and SI. Data is reused from <https://doi.org/10.1101/2021.04.14.439776>, with permission from the authors

- In the discussion authors should reconcile their observations with previous works by the Sabatini and Yilmaz groups showing that calorie restriction mediated by reduced mTOR pathway activity leads to Lgr5+ ISCs increased number due to a bystander effect on Paneth cells?

This has been added to the text. The Yilmaz publications have shown that the response to calorie restriction is mediated via the Paneth cell. In our organoid model we lose Paneth cells (Fig. 1D, Supplementary Fig. 1B), preventing them from influencing the ISC response.

- It is surprising that Zaka KO or knockdown per se does not lead to any phenotype. If ribosome stalling and collisions have an important physiological role in sensing faulty mRNAs, Zaka modulation should produce some type of cellular phenotype. Are there alternative pathways mediating this process? This should be discussed.

This is a very interesting comment, **expand** and we have updated the manuscript to reflect this point.

- Authors do not show how collision-induced signaling through Zaka specifically targets glycolysis-related genes in ISCs only.

This is obviously a question that we are very interested in, and one we have not quite answered. We have now carried out Zaka IP, and combined this with mass spec to understand how Zaka is regulating the phenotype. This experiment has revealed a Zaka-Src-Yap signaling cascade that is required for the adult to fetal switch that we have described. This however, activates a transcriptional response, which does not explain how metabolism is regulated by translation. However, the proteomic data has some interesting targets that may play a role. For example, Zaka appears to bind to Btf3 following Rptor deletion. As Btf3 is known to drive the localization of ribosomes to the mitochondria, that presents the possibility that such localization may play a role, as has been previously shown for mRNAs encoding for oxidative phosphorylation related mRNAs (e.g. PMID: 31929983, PMID: 26151724).

As a second approach, we have analyzed the features of the mRNAs that are differentially translated, and have not identified a consistent pattern that could explain their specific regulation.

Reviewer #2 (Remarks to the Author):

Silva et al. presented exciting and captivating observations in which Zaka-dependent, yet Gcn2-elf2 α -independent, pathway results in ISCs to switch cell identity. Given that growing understanding of the ribosome collision as a sensor for cellular status, the finding of the involvement in intestinal stem cell identity is quite timely and will appeal to board readers. However, the discussion about ribosome profiling data and ribosome collision was not helpful in

the authors' reasoning. In addition, the direct detection of ribosome collisions was lacking. The following concerns are needed to be addressed before publication.

We would like to thank the Reviewer for their time and their kind comments.

Major comments:

1. There was no direct evidence linking ribosome collisions to the observed phenotype. Although authors speculated that all the conditions (Raptor depletion, low concentration of cycloheximide, and depletion of amino acids) may incur ribosome collisions, no direct evidence of ribosome collision was provided. It is necessary to show solid verification of ribosome collisions by, for example, sucrose density gradient followed by RNase digestion to probe the increase of RNase-resistant disomes. Sequencing of the disome footprints (i.e., disome profiling/disome-seq) could be an alternative approach.

This is a very valid point, and something we have struggled with. Unfortunately organoids are difficult and expensive to work with, and despite our best efforts, we simply cannot gather enough material to carry out the suggested experiment (RNase digestion followed by sucrose gradient).

We also tried disome profiling following the Reviewer's suggestion, but encountered the same issues. While we could get sufficient sequencing depth in Rptor KO organoids (~900k reads), we could not get enough in a WT context (~50k reads). This prevents comparative analysis. It is tempting to suggest that this is as a result of increased collisions following Rptor deletion, however we do not have evidence to back this up.

As a result, we have tempered our language throughout the manuscript, and now refer to "ribosome impairment" rather than "collisions". While we do believe that collisions play a role (there are periodic peaks preceding the stop codon in the disome seq data that are indicative of ribosome queuing), we do not have evidence at present to make this claim.

2. Fig. 2G showed that footprint accumulation on the stop codon. First, this data was difficult to interpret. The authors should consider another way of data visualization. Second, the data suggested the accumulation of footprints that hold stop codons at P site, in addition those at A site. This seemed to be unusual since the ribosome should be dissociated from mRNA where the A site lies at the stop codon. Ribosomes on P-site stop codon may represent the post-termination, unrecycled ones. The author should carefully characterize the footprints.

We thank the reviewer for their comment. These are indeed very difficult graphs to interpret. First, with regard to the apparent P-site accumulation, we have changed the graph to show RPF abundance (as a percentage) as well as Log2FC. The FC measure of P-site stop codon residence is indeed bigger than that in the A-site, but the abundance of reads is significantly lower. So in real terms, there are far more reads with the stop codon in the A-site than the P-site. Although we do still see some reads with the stop codon in the P-site, we believe that this is probably due to poor EPA site characterization.

With respect to visualization of the data, we have tried a number of different approaches, and have settled on what we present in the revised manuscript. It is still a difficult graph to follow, so we have included a more thorough explanation in the Figure legend.

3. Related to the point above, the ribosome pausing on the stop codon was not supported by the meta-gene analysis in Supplemental Fig 4C.

Again, this is an issue with our explanation. We would like to clarify that the meta-gene analysis does not visualize the read abundance at codon-level resolution. The CDS-end position in the X-axis represents a certain percentage of positions at CDS ends, therefore corresponds to a bin with the number of codons dependent on the length of the mRNA.

As mentioned above, we have updated the Fig. 2C to plot the RPF abundances at codon-level resolution, with the hope that this more clearly illustrates our point.

4. Not simply focusing on the stop codon, a more comprehensive survey of ribosome pauses induced by Raptor depletion should be carefully conducted (e.g., codon/amino acid features associated with the pausing).

We have added this analysis to the Supplementary material (Supplementary Fig. 6). It shows that the pausing is specific to the Stop codon.

5. Regarding Raptor depletion and inactivation of mTORC1, the most straightforward impact in translation would be reduced translation initiation, mediated by LARP1 and/or 4EBP1 (especially, dephosphorylated LARP1 would lead to the strong repression of ribosome protein translation.) However, the contribution of this effect was not discussed in the manuscript (regarding the phosphorylation status and translation efficiency changes in ribosome profiling/RNA-seq data). Authors may speculate the larger impact on elongation (such as reported in authors' earlier work of Faller et al. Nature 2015) through eEF2 phosphorylation. However, this point was not discussed in the manuscript. Again, the effect on the termination/ribosome recycling as described in Fig. 2G was not linked to either initiation or elongation scenario. The authors were requested to provide a reasonable explanation for their observations on inefficient termination/recycling by Raptor depletion.

This is a good suggestion, and one we have thought about. We do not believe this to be a direct result of the inhibition of mTOR however. RL Fig. 4 (below) shows that conditions that cause the adult to fetal switch (short term glutamine deprivation) do not cause a decrease in mTORC1 signaling, as measured by S6 phosphorylation. To assess the role of Larp1 in this context, we analyzed the translation of TOP-containing mRNAs. This revealed a slight decrease in the TE of these mRNAs, largely driven by decreased TE of ribosomal proteins. While the effect was subtle, it was statistically significant. It is difficult to assign a causative role to such changes however, and we do not believe that it will significantly add to the manuscript.

In response to the Reviewer's request to provide an explanation of the apparent inefficient termination/recycling, we do see a decrease in the expression of several factors related to translation termination (eRF3, Abce1, and Denr). This could explain the change in stop codon

occupancy, but as we have not probed this in greater detail, we are not sufficiently confident of this observation to include it in the manuscript.

RL Fig. 4: Phosphorylation of S6 is not decreased after glutamine deprivation.

Minor comments:

1. The TE of glycolysis-related genes increased upon Raptor depletion. Yet, the author should provide evidence at the protein level. Western blot or mass spectrometry would be helpful to strengthen the authors' conclusion. The same also applied to oxidative phosphorylation-related genes.

We would like to point out that we have a functional readout of both oxidative phosphorylation and glycolysis (Seahorse analysis), and we believe that provides a much stronger confirmation of altered pathway translation than a Western blot. We have also now included a volcano plot showing the TE changes in both oxidative phosphorylation and glycolysis-related mRNAs (Supplementary Fig. 3B).

2. In line 66, although this reviewer understood that the use of phrase “the first evidence” was intended to stress the importance of the findings, more objective phrases are recommended, since such a wording may not fit to the journal policy.

We have altered this in the text.

3. In line 123, the citation of paper 41 at the phrase of “Upon further analysis of this data⁴¹,” was not clear whether this indicated the data analysis of earlier paper or that obtained in this study. Please clarify this.

This has been updated.

4. The detail for the RNA-seq experiments was not described in the method section. This reviewer imagined that the experiments were conducted in whole organoid not in HA-tag

isolated ribosome fraction as done in ribosome profiling, however, the details should be described.

We apologize for this oversight, and thank the Reviewer for picking it up. We have updated the Methods section of the manuscript with the relevant details.

5. Related to this, mRNA areas used for read counting for ribosome profiling and RNA-seq were not clear in the method. Given that ribosome footprint originates mainly from ORF, the read counting of RNA-seq should be also restricted to the ORF for fair comparison. Please clarify this point in the method. If the entire mRNA region was used for the read counting, the reanalysis of the data is recommended.

We updated the relevant section to clarify the methodology used. For differential TE analysis, we do not define the Translation Efficiency as the direct read ratio between Ribo-seq and RNAseq reads. We define it as the ratio between the ribosome occupancy and the abundance of a transcript. Therefore, our analysis is not directly performed using the ORF read-counts. Instead, we quantify the mRNA abundances using bioinformatic tools that correct for commonly observed biases and handle the reads fairly when not uniquely mapped. As reads from UTRs are by default present in RNAseq data, it is not fair to exclude them for accurate mRNA abundance quantifications. However, since no (or minimal) Ribo-seq reads are expected at UTRs, mRNA sequences are trimmed for the quantification of ribosome occupancy.

Reviewer #3 (Remarks to the Author):

General comments:

In this manuscript, Silva and colleagues examine the effect of translation inhibition on the identity of intestinal stem cells (ISCs). They make the interesting claim that by activating Zaka, ribosome collisions drive a change in ISCs to a fetal-like identity. In the first initial experiments, the authors took advantage of organoids with a conditional knock out of Raptor to inhibit mTORC1. Upon the loss of Raptor, these cells reverted to a fetal-like identity and, more importantly, rewired their metabolism from oxidative phosphorylation to glycolysis. This switch appears to be regulated through translation and not transcription. The authors then went to show ribosome stalling through the addition of cycloheximide results in a similar phenotype and make the conclusion that it is likely to be collisions since addition of cycloheximide to high concentrations does not trigger this switch. This then led the authors to examine the role of Zaka, which has recently been shown to be activated by ribosome collisions, in the process. The authors showed that the factor is important for the switch in the identity, as well as metabolic rewiring, but that its canonical function as a MAPKKK is dispensable for this activity.

These studies expand on the newly identified roles of ribosome collisions on triggering diverse signaling pathways, and as such, its subject is topical and relevant to a wide audience. However, the manuscript in its current form lacks important experiments and controls to support the bold conclusions made by the authors. Indeed, the authors make the argument that collisions are responsible for this switch using one experiment – they show that low concentrations of cycloheximide also drive the switch. However, there was no effort to show that collisions took place under these conditions. Furthermore, the authors did not attempt to investigate the mechanism by which Zaka activation results in changes in translation efficiency.

Specific comments

There are several obvious experiments that the authors can do to bolster their arguments/conclusions:

1- The authors need to show that deletion of Raptor results in ribosome collisions; currently this is all based on conjecture. They also need to show that cycloheximide addition at the concentration used in their studies results in ribosome collisions.

As we have already said in reply to Reviewer 2, this is something we have struggled with. Unfortunately organoids are difficult and expensive to work with, and despite our best efforts, we simply cannot gather enough material to carry out the suggested experiment (RNase digestion followed by sucrose gradient).

We also tried disome profiling following Reviewer 2's suggestion, but encountered the same issues. While we could get sufficient sequencing depth in Raptor KO organoids (~900k reads), we could not get enough in a WT context (~50k reads). This prevents comparative analysis. It is tempting to suggest that this is as a result of increased collisions following Raptor deletion, however we do not have evidence to back this up.

As a result, we have tempered our language throughout the manuscript, and now refer to "ribosome impairment" rather than "collisions". While we do believe that collisions play a role (there are periodic peaks preceding the stop codon in the disome seq data that are indicative of ribosome queuing), we do not have evidence at present to make this claim.

2- How does the activation of Zaka affect ribosome function? In other words, the factor is a MAPKKK, but the authors argue that this function is unrelated to what they observe here. So how does it affect the ribosome's ability to translate certain genes, but not others?

This is a very pertinent question, and one we have expanded on in this revision. The first thing to say is that while Zaka function is independent of its MAPKKK activity, it still appears to be

dependent on its kinase activity. We have now shown that the Zaka-dependent phenotypic switch is as a result of the phosphorylation and activation of Src, which in turn activates Yap. This signaling cascade is required for the switch in ISC identity, and explains the role of Zaka in this context.

The second part of the Reviewer's question is to do with the specificity of translation that we observe. We have analyzed the features of the mRNAs that are differentially translated, and have not identified a consistent pattern that could explain this. However, the proteomic data that we have added to the manuscript has some interesting targets that may play a role. For example, Zaka appears to bind to Btf3 following Rptor deletion. As Btf3 is known to drive the localization of ribosomes to the mitochondria, that presents the possibility that such localization may play a role, as has been previously shown for mRNAs encoding for oxidative phosphorylation related mRNAs (e.g. PMID: 31929983, PMID: 26151724).

3- The authors only used cycloheximide as a probe for ribosome collision. Why only cycloheximide, especially given that usage of the drug has many caveats and others in the ribosome-collision field use other drugs? To that end, amino acid starvation does not only trigger ribosome collisions, but it also inhibits mTORC1, so this becomes a circular argument, leading us back to the effects seen in Figures 1 and 2 in the absence of Raptor.

Following the Reviewer's suggestion, we treated the organoids with either anisomycin or emetine. These treatments gave inconclusive results. As collisions caused by different insults can have very different consequences (e.g. PMID: 35180429), this is not incompatible with collisions causing the observed phenotype. However, it is enough to cause us to temper our language. We have therefore removed our assertion that this is collision related, and instead describe it as being the result of "ribosome impairment".

In answer to the second part of the Reviewer's comment regarding the inhibition of mTORC1, we do not believe this to be the root cause of the phenotypic switch. As can be seen above in RL Fig. 4, deprivation of glutamine is not sufficient to inhibit mTORC1 signaling in this context, even though there is a robust and reproducible phenotypic switch. Furthermore, we do not see an inhibition of mTORC1 following low dose cycloheximide treatment. Instead, we believe that each of these insults causes a ribosome impairment that results in the activation of Zaka.

4- The authors need to convincingly show that GCN2 can be activated in their system using other stressors.

We thank the Reviewer for this question, but are not clear on how it would add to the manuscript. Gcn2 has been studied previously in the intestine (PMID: 26982722, PMID: 31685988), and there is no suggestion that it cannot be activated in this tissue. Furthermore, we do not see why it would be an issue if it could not be activated. We show that it does not play a role in detecting glutamine or leucine deprivation in our system; if it could not be activated then this would further support our findings, not undermine them.

5- The phosgel in Figure 3C is not convincing. The same gels used by the Green group show a much greater shift for Zaka when phosphorylated.

We have not used phospho-gels in our manuscript. We have used the protocol published in PMID: 35316659 and PMID: 32289254, where a band shift of a similar magnitude was observed. We have included a western of our FLAG-tagged Zaka (RL Fig. 2), as an additional confirmation of our findings.

REVIEWERS' COMMENTS

Reviewer #1 (Remarks to the Author):

The authors have satisfactorily addressed the points raised by this reviewer.

Reviewer #2 (Remarks to the Author):

All of my concerns were addressed by the authors. This report definitely adds more evidence for nutrient sensing mechanisms during translation. The new data for Zaka-SRC-YAP signaling was valuable. This reviewer has a only minor suggestion; regarding unshown data for disome profiling and western blot of eRF1/ABCE1, it is worth considering to include those data in the supplemental figures, if possible.

Reviewer #3 (Remarks to the Author):

In the revised manuscript, the authors appear to have conducted additional experiments to address the reviewers' concerns. They also rewrote the manuscript and added new text to make the manuscript more streamlined. And while I recognize these efforts, the authors new data seem to be inconsistent with ribosome collisions playing a role in intestinal stem cell identity. Indeed, the authors changed the title of the manuscript and rewrote parts of the paper to temper their earlier conclusions about the role of ribosome collisions in the process. I should note that I appreciate the authors' willingness to reframe their interpretation. However, the fact remains that the role of ribosome impairment/stalling in this process hinges on one piece of data (the cycloheximide one). Interestingly, the authors mentioned that anisomycin and emetine gave inconclusive results, but anisomycin is the drug of choice for activating Zaka α and the ribosome stress response (RSR).

Specific comments:

1) Can the authors comment on how much RNA can be obtained from organoids? For other systems, absorbance traces of sucrose-gradient fractionations can be obtained using 50-100 ug of total RNA to analyze polysome compositions.

2) It is unclear how many times the experiments were repeated for some of the blots. For example, the bar graph in Figure 4C appears to have been conducted only once.

REVIEWERS' COMMENTS

Reviewer #1 (Remarks to the Author):

The authors have satisfactorily addressed the points raised by this reviewer.

We thank the reviewer for their time and valuable input.

Reviewer #2 (Remarks to the Author):

All of my concerns were addressed by the authors. This report definitely adds more evidence for nutrient sensing mechanisms during translation. The new data for Zak α -SRC-YAP signaling was valuable. This reviewer has a only minor suggestion; regarding unshown data for disome profiling and western blot of eRF1/ABCE1, it is worth considering to include those data in the supplemental figures, if possible.

We thank the reviewer for their time and valuable input.

With regard to the reviewer's suggestions, we would not be comfortable publishing our disome seq. As we mentioned in our last rebuttal letter, the differences we see in that data can be explained by an increase in ribosome collisions. However, there are also technical explanations for the data, and we would not trust our interpretation without additional confirmation. While we believe our interpretation is correct, we would not be happy publishing it.

Reviewer #3 (Remarks to the Author):

In the revised manuscript, the authors appear to have conducted additional experiments to address the reviewers' concerns. They also rewrote the manuscript and added new text to make the manuscript more streamlined. And while I recognize these efforts, the authors new data seem to be inconsistent with ribosome collisions playing a role in intestinal stem cell identity. Indeed, the authors changed the title of the manuscript and rewrote parts of the paper to temper their earlier conclusions about the role of ribosome collisions in the process. I should note that I appreciate the authors' willingness to reframe their interpretation. However, the fact remains that the role of ribosome impairment/stalling in this process hinges on one piece of data (the cycloheximide one). Interestingly, the authors mentioned that anisomycin and emetine gave inconclusive results, but anisomycin is the drug of choice for activating Zak α and the ribosome stress response (RSR).

We thank the reviewer for their time and valuable input. However we have to slightly take issue with their assertion that "the role of ribosome impairment/stalling in this process hinges on one piece of data (the cycloheximide one)". We also have evidence of stalling on the stop codon following Rptor deletion, which is a clear indicator of ribosome impairment (Fig. 2G). Alongside this we see a substantial decrease

in 35S methionine incorporation (Supp. Fig. 7A). Taken together with the Cycloheximide data, we believe that this provides sufficient evidence of ribosome impairment.

Specific comments:

1) Can the authors comment on how much RNA can be obtained from organoids? For other systems, absorbance traces of sucrose-gradient fractionations can be obtained using 50-100 ug of total RNA to analyze polysome compositions.

Using Trizol-based RNA extraction, the absolute maximum amount of RNA we can get roughly is 400ng from one confluent matrigel plug. To reach the amount of RNA needed for visualization, we would need at least 125 plugs. However, Trizol is not compatible with sucrose gradients, so we have to use a less efficient extraction method, increasing the number of plugs required. We did try extensively to generate absorbance traces (up to 200 plugs!), but without success.

2) It is unclear how many times the experiments were repeated for some of the blots. For example, the bar graph in Figure 4C appears to have been conducted only once.

The figure legends have been updated to include this data in a clearer manner.